# LARGE-SCALE PRETRAINING OFFERS MODEST BENEFITS FOR TABULAR TRANSFER LEARNING

## ABSTRACT

Several recent works seek to train foundation models for tabular prediction by pretraining neural networks on large collections of tabular classification and regression datasets. These *tabular foundation models* (TFMs) are often reported to outperform non-pretrained baselines when applied to predictive tasks on unseen tables, demonstrating effective tabular transfer learning. In this paper, we show that, in contrast to the positive conclusions of prior works, the perceived performance benefits from large-scale tabular pretraining largely diminish when we aggregate the results across datasets while (i) preserving the performance differences between models in their *original scale* (e.g., without min-max normalization); and (ii) testing for the statistical significance of these differences. For example, when we replicate the original evaluation setup for TabPFN-v2 on classification tasks, TabPFN-v2 indeed achieves the highest average min-max normalized AUROC, but reaches a statistical tie with CatBoost in 69% of all datasets, while significantly outperforming it in 20.7% of datasets and underperforming it in the remaining 10.3% of datasets. We evaluate seven open-source TFMs on 88 classification and 82 regression datasets in both full-data (i.e., using all training examples) and few-shot settings, and find that existing TFMs only show statistically significant improvements over non-pretrained baselines on small classification datasets, with no consistent gains in other settings. To isolate the impact of tabular pretraining, we also compare three TFMs directly to their non-pretrained counterparts, and find that, in most cases, the performance gains from pretraining are minimal. Our findings suggest that current approaches to large-scale tabular pretraining may offer limited performance benefits, showing room for improvement in methods for effective tabular transfer learning. For standardized and reproducible evaluation of TFMs, we release our evaluation suite as the *TFM Evaluation Harness*.

## 1 INTRODUCTION

Tables are one of the most ubiquitous forms of data across various real-world domains, including electronic health records in medicine (Pollard et al., 2018; Johnson et al., 2016), risk assessment records in criminal justice (Larson et al., 2016), and survey data in the social sciences (Kohavi, 1996). Owing to their heterogeneity, they remain one of the most challenging modalities for machine learning (ML), with gradient-boosted decision trees (GBDTs; Friedman, 2001)—such as XGBoost (Chen & Guestrin, 2016) and CatBoost (Prokhorenkova et al., 2018)—maintaining their edge over modern approaches based on deep neural networks. Prior works on deep tabular models propose to improve performance by e.g., designing specialized architectures (Huang et al., 2020; Arik & Pfister, 2021); performing extensive hyperparameter tuning (Kadra et al., 2021); improving the encoding of numerical features (Gorishniy et al., 2022); and pretraining with self-supervision (Somepalli et al., 2021; Bahri et al., 2022)—many of which conclude that the proposed method outperforms GBDTs with state-of-the-art results. However, follow-up works often reveal that such positive conclusions are not robust (Shwartz-Ziv & Armon, 2021; Grinsztajn et al., 2022; McElfresh et al., 2023; Ye et al., 2025a), often still leaving GBDTs as the go-to approach in various practical settings.

Meanwhile, recent advances in foundation models for images and text have spurred significant interest in the development of *tabular foundation models* (TFMs), with the hopes that *large-scale tabular pretraining* would unlock similar benefits, enabling deep tabular models to outperform traditional baselines via cross-table transfer learning (Wang & Sun, 2022; Levin et al., 2023). Some of the most successful examples of TFMs include (i) tabular in-context learning (ICL) models pretrained to

implement approximate Bayesian inference in a single forward pass (Hollmann et al., 2023; 2025; Ma et al., 2024; Qu et al., 2025); and (ii) large language model (LLM)-based models, which aim to leverage their parametric knowledge and language capabilities to improve cross-table transfer learning (Wang et al., 2024; Kim et al., 2024; Yan et al., 2024; Gardner et al., 2024). By evaluating on new, unseen tables, **all such works claim significant improvements over non-pretrained baselines**, especially on small-sized datasets (Hollmann et al., 2025; Qu et al., 2025; Erickson et al., 2025).

Despite such positive conclusions, it is worth scrutinizing (i) how generalizable these findings are and (ii) how much of the performance improvements can truly be attributed to tabular pretraining, as the evaluation setups vary significantly. For example, TabDPT (Ma et al., 2024), CARTE (Kim et al., 2024), and TP-BERTa (Yan et al., 2024) are all evaluated on datasets with different inclusion criteria, each restricted to datasets with particular characteristics, but applied inconsistently; and TabPFN-v2 (Hollmann et al., 2025) internally performs additional feature preprocessing (e.g., outlier removal, power transforms), while the baselines may not undergo comparable preprocessing. All such factors can impact the conclusions about the effectiveness of each tabular pretraining approach (Tschalzev et al., 2024), calling for a standardized protocol that ensures a fair and rigorous assessment of TFMs.

In this paper, we stress-test seven open-source TFMs on 88 classification and 82 regression datasets in both full-data (i.e., using all training examples) and few-shot settings while addressing these issues, and find that **TFMs show modest improvements in narrow settings but generally fail to deliver consistent, statistically significant gains over non-pretrained baselines (Section 4)**. In surfacing this finding, we identify a critical gap in common evaluation approaches in tabular ML: comparing models based on *min-max normalized* evaluation metrics and ranking-based hypothesis tests may fail to capture *whether the absolute gains in performance themselves are statistically significant*, potentially leading to overly optimistic conclusions about the effectiveness of tabular pretraining.

Our conclusions follow from testing the statistical significance of absolute performance gains (*on the original metric scale*) of TFMs over the best-performing baselines via the percentile bootstrap, and aggregating these pairwise results into a coherent ranking with the Elo system (Elo, 1978) (Section 3). We also perform an *apples-to-apples* comparison of three TFMs vs. their non-pretrained counterparts to isolate the performance benefits from large-scale pretraining itself, where we observe modest gains that do not consistently translate into superior performance over baselines. Our findings suggest that current approaches to large-scale tabular pretraining may offer limited performance benefits, showing room for improvement in methods for effective tabular transfer learning. Our main contributions are:

1. We demonstrate that evaluating TFMs based on min-max normalized evaluation metrics and ranking-based hypothesis tests may fail to capture the statistical significance of any measured performance gain, potentially leading to overly optimistic conclusions.

2. We stress-test seven open-source TFMs on 88 classification and 82 regression datasets, carefully curated to cover a diversity of real-world settings, and find that TFMs generally fail to consistently show statistically significant performance gains over baselines.

3. We compare three TFMs directly against their non-pretrained counterparts to fully isolate the impact of large-scale tabular pretraining on downstream performance, and find that the performance improvements are limited and fail to surpass non-pretrained baselines.

## 2  RELATED WORKS

**Deep neural networks vs. GBDTs.**  Deep learning models for tabular data have historically lagged behind GBDTs (Friedman, 2001)—such as CatBoost (Prokhorenkova et al., 2018), XGBoost (Chen & Guestrin, 2016), and LightGBM (Ke et al., 2017)—which have effective inductive biases for learning on heterogeneous structured data (Grinsztajn et al., 2022; McElfresh et al., 2023). Over the years, various deep learning approaches have been proposed to improve over GBDTs, including but not limited to tree-inspired neural network architectures (Lay et al., 2018; Popov et al., 2020), specialized numerical and categorical feature encoding methods (Gorishniy et al., 2022; Holzmüller et al., 2024), rigorous regularization techniques (Kadra et al., 2021), self-supervised pretraining (Yoon et al., 2020; Somepalli et al., 2021; Bahri et al., 2022), kNN-style retrieval (Gorishniy et al., 2024; Ye et al., 2025b), and deep ensembling (Gorishniy et al., 2025). However, follow-up studies often suggest that the claimed improvements over GBDTs are limited (Shwartz-Ziv & Armon, 2021; Grinsztajn et al., 2022; Ye et al., 2025a), and while both model families have complementary strengths (McElfresh et al., 2023), achieving a similar level of performance with neural networks often requires extensive model training and hyperparameter tuning, which can be impractical in many real-world settings.

**Tabular transfer learning and foundation models.**   Motivated by the successes of transfer learning in vision and language, several recent works in tabular ML propose to transfer "knowledge" *across tables* by pretraining or jointly training on a large collection of tables. Early works on cross-table transfer learning demonstrate that when a model is pretrained on tables semantically related to the downstream table of interest with highly overlapping columns, its downstream performance can improve significantly (Wang & Sun, 2022; Levin et al., 2023). With the advent of LLMs and more flexible architectures (e.g., Transformers (Vaswani et al., 2017)) that are capable of handling multiple tables with varying structure, more recent works propose to pretrain models on a large miscellaneous collection of tabular datasets, resulting in a foundation model for tabular prediction (i.e., TFM).

Some of the most empirically successful examples of TFMs include (i) tabular ICL models, such as TabPFN (Hollmann et al., 2023; 2025), TabICL (Qu et al., 2025), and TabDPT (Ma et al., 2024); and (ii) LLM-based models, such as TabuLa-8B (Gardner et al., 2024), TP-BERTa (Yan et al., 2024), and CARTE (Kim et al., 2024), while other examples such as XTab (Zhu et al., 2023) and UniTabE (Yang et al., 2024) also exist. Tabular ICL models are often pretrained such that, given a sequence of tokens corresponding to the training examples followed by some test examples, a forward pass of the model implements approximate posterior predictive inference with respect to a prior over plausible tabular datasets (Müller et al., 2022). Prior works show that tabular ICL models often achieve strong performance on small-scale datasets (McElfresh et al., 2023; Zabërgja et al., 2025; Erickson et al., 2025), with improved robustness to common data challenges such as presence of missing values and/or uninformative features (Hollmann et al., 2025). LLM-based TFMs are often shown to show strong performance in limited-data settings and on datasets with rich textual semantics (Gardner et al., 2024; Yan et al., 2024; Kim et al., 2024), albeit with limited applicability to large datasets that contain many samples and features due to inefficient tokenization and high computational cost.

## 3   EVALUATION APPROACH

To investigate the performance benefits from large-scale tabular pretraining, we stress-test seven open-source TFMs against baselines on 88 classification and 82 regression tasks, in both the full-data (i.e., using all training examples) and few-shot learning settings. In the full-data setting, we assume that all of the training examples are available for model training or ICL. In the few-shot setting, on the other hand, we randomly subsample $k = 1, 2, 4, 8, 16, 32, 64, 128, 256, 512, 1024$ examples (*per class*, for classification tasks) from the training set and limit the access to just those examples. As such, for models that need to be trained, we further split the $k$-shot examples into a training set and a validation set. When $k < 4$, we take a (stratified) 50–50 split; otherwise, we take a 75–25 train–validation split. For ICL models, we provide all of the subsampled examples as context for prediction. To account for the randomness in subsampling, we always repeat each experiment five times with different random seeds and report the average performance, for each model and dataset in the $k$-shot setting. For the classification tasks, we use the area under the receiver operating characteristic curve (AUROC) as the main metric, and take a micro-average in multi-class settings. For the regression datasets, we use the root mean squared error (RMSE) and the coefficient of determination ($R^2$) as the main metrics.

**Benchmark datasets.**   To construct the evaluation suite, we first combine six widely used tabular prediction benchmarks—TALENT (Ye et al., 2025a), TabZilla (McElfresh et al., 2023), Grinsztajn et al. (2022), AutoML Benchmark (AMLB; Gijsbers et al., 2024), OpenML-CC18 (Vanschoren et al., 2014), and OpenML-CTR23 (Fischer et al., 2023)—to construct a candidate pool of 163 classification and 105 regression datasets from OpenML (Vanschoren et al., 2014). We perform this aggregation to ensure that the datasets considered for evaluation are not biased towards a particular setting (e.g., Grinsztajn et al. (2022) focus on datasets with $\leq$ 10k samples, no missing values, and low-cardinality categorical features), which is critical for drawing robust and generalizable conclusions (Kohli et al., 2024). We then apply the following exclusion criteria in the order presented:

1. Exclude "non-tabular" datasets (e.g., Fashion-MNIST (Xiao et al., 2017) in OpenML-CC18);
2. Exclude datasets used to pretrain any of the TFMs (e.g., TabZilla datasets for TabDPT);
3. Exclude datasets that are "too easy" to reflect real-world settings.

For Step 1, we consider a dataset to be "non-tabular" if its most natural representation is *not* in the form of a table, considering its content (Kohli et al., 2024)[1]. For Step 3, we consider a classification dataset as "too easy" if *either* a logistic regression *or* a kNN model achieves $\geq$ 0.95 test AUROC,

---

[1]We still include datasets that are originally from "non-tabular" data but contain tabular summary statistics.

without any hyperparameter tuning (i.e., using the default hyperparameters in `scikit-learn`). Similarly, for regression, we check if either a Ridge regression or a kNN model achieves $\geq 0.95$ test $R^2$. This is in contrast to benchmarks like TALENT (Ye et al., 2025a), where the "easy" datasets (based on a different definition) are deliberately retained for evaluation. Otherwise, we do not perform any additional filtering based on dataset size, number/ratio of numerical vs. categorical features, missingness, class imbalance, collinearity, etc. to cover a diversity of challenging settings in the wild. We also assume the standard i.i.d. setting and do not consider any form of distribution shift (as in e.g., Gardner et al. (2023); Rubachev et al. (2024)), which we leave as future work.

To prepare each dataset for training and evaluation, we use the official 90–10 train–test splits provided through OpenML, following standard practice (McElfresh et al., 2023; Gijsbers et al., 2024; Zabërgja et al., 2025). As we found that several datasets have train–test leakage (i.e., exactly same features and target label) upon close inspection, we first remove all such instances from the test set. We then correct any mislabeling of the features (e.g., numerical features labeled as categorical and vice versa) via an automated correction criteria similar to Hollmann et al. (2025)[2]. We also remove features that exhibit zero variance in the training set or correspond to sample identifiers (e.g., patient IDs). We convert any timestamps (e.g., `2025-01-16 12:00:00`) into a set of sinusoidal numerical features, following Hoo et al. (2024). For all models except TabuLa-8B and Llama-3-8B, we then $z$-score normalize all numerical features and one-hot encode all categorical features before providing them as input. For missing values, we only perform imputation (mean/mode imputation for numerical/categorical features) if a model is incapable of handling them. Meanwhile, for regression tasks, we do not transform the target variables in any way, unlike often done in prior works to handle heavy-tailed target distributions (e.g., log-transforms). We provide the remaining details in Appendix A.

**Tabular foundation models (TFMs).** We evaluate three tabular ICL models—TabPFN-v2 (Hollmann et al., 2025), TabICL (Qu et al., 2025), and TabDPT (Ma et al., 2024); three LLM-based models—TabuLa-8B (Gardner et al., 2024), TP-BERTa (Yan et al., 2024), and CARTE (Kim et al., 2024); and XTab (Zhu et al., 2023), a Transformer model pretrained on a subset of AMLB datasets. TabICL and TabuLa-8B are limited to classification, while other TFMs can be used for both classification and regression. Meanwhile, TabPFN-v2 can be evaluated with post-hoc ensembling (PHE; detailed in the "TabPFN (PHE)" section of Hollmann et al. (2025)), where each prediction is obtained by aggregating the outputs from *multiple* instantiations of TabPFN-v2 with greedy ensemble selection (Caruana et al., 2004). However, *we intentionally exclude PHE* for TabPFN-v2 to (i) ensure a consistent setup with other TFMs which are not post-hoc ensembled in the original paper and (ii) isolate the impact of tabular pretraining on downstream performance as much as possible. **The "default" preprocessing steps and inference configuration for TabPFN-v2 (see Extended Data Table 5 of Hollmann et al. (2025)) are still internally applied within the model's forward pass.**

For all tabular ICL models, whenever the total number of training examples exceeds the context window size, we retrieve the training examples with the smallest $L_2$ distance (in the normalized feature space) to each test example for ICL (Thomas et al., 2024; Ma et al., 2024). For TabuLa-8B, fine-tuned from Llama-3-8B (Grattafiori et al., 2024), we only evaluate it in the few-shot setting, as its context window size is too small to fit entire datasets. When generating predictions from TabuLa-8B and Llama-3-8B, we follow Hegselmann et al. (2023) and treat the conditional log-probability of the token sequence corresponding to each class, normalized over all classes, to be the model's confidence score. We treat the prompt format for Llama-3-8B as a hyperparameter and optimize it based on validation AUROC (when $k \geq 2$), following Jeong et al. (2024) (Appendix B.1). For TP-BERTa, which was adapted from RoBERTa (Liu et al., 2019) for tabular prediction, we use the checkpoint pretrained for each prediction task (i.e., classification or regression), as Yan et al. (2024) show that pretraining on both tasks leads to worse performance on each task. We denote the model pretrained on each task as TP-BERTa (CLF) and TP-BERTa (REG). For XTab, we use the checkpoint with a FT-Transformer (Gorishniy et al., 2021) backbone. We provide the remaining details in Appendix B.

**Baselines.** For comparison, we consider six neural network baselines: TabR (Gorishniy et al., 2024), ModernNCA (Ye et al., 2025b), MLP-PLR (Gorishniy et al., 2022), FT-Transformer (FT-T; Gorishniy et al., 2021), ResNet (Gorishniy et al., 2021), and MLP; and seven classical baselines: CatBoost (CB; Prokhorenkova et al., 2018), LightGBM (LGBM; Ke et al., 2017), XGBoost (XGB; Chen & Guestrin, 2016), random forest (RF; Breiman, 2001), SVM (Cortes & Vapnik, 1995), kNN,

---

[2]See e.g., https://github.com/PriorLabs/TabPFN/blob/9a4d401/src/tabpfn/utils.py#L520

and linear/logistic regression (LR). We focus on baselines that generally perform well according to recent benchmark studies (McElfresh et al., 2023; Ye et al., 2025a; Erickson et al., 2025) and leave the inclusion of other relevant models as future work. We also consider the non-pretrained versions of TFMs (if applicable) as additional baselines: Llama-3-8B (Grattafiori et al., 2024) for TabuLa-8B, the random and RoBERTa initializations for TP-BERTa, and the random initialization for XTab.

**Training and hyperparameter optimization.** To ensure we compare all models based on their best performances, we extensively optimize the hyperparameters of each model we train based on their validation AUROC (or RMSE), searching through 100 hyperparameter configurations sampled via a Tree-structured Parzen Estimator (TPE; Watanabe, 2023) in `Optuna` (Akiba et al., 2019). One exception we make is for TP-BERTa, for which we sample 30 hyperparameter configurations, due to the relatively high runtimes associated with fine-tuning a 125M-parameter model. Unlike many prior works (McElfresh et al., 2023; Hollmann et al., 2025; Zabërgja et al., 2025; Erickson et al., 2025), we do not impose any timeout on the hyperparameter optimization process, so as not to confound the interpretation of results. Within each trial, we train each model for a maximum of 200 epochs, with an early stopping patience of 10 epochs. In the few-shot setting, we only optimize the hyperparameters when the number of few-shot examples $k \geq 8$, given that the validation performance may be an unreliable measure of generalization given too few examples. For the classical baselines, we tune the hyperparameters based on their 4-fold average cross-validation performance. For the neural network baselines, given that full-blown cross-validation can be computationally costly, we take a random 75–25 train–validation split, and then use the performance on the resulting validation set for model selection. We run the classical baselines on CPUs only, and the neural network baselines on a single 48GB NVIDIA L40S or A6000 GPU for each trial. We include the remaining details in Appendix B.

**Aggregating the results for model comparison.** When comparing ML models based on their performance on multiple datasets, popular approaches are to (i) average the min-max normalized evaluation metric (Grinsztajn et al., 2022) and (ii) perform ranking-based pairwise hypothesis testing (Demšar, 2006). For the former, the test scores of all models on each dataset are projected onto a [0,1] scale, with the worst-performing model receiving a score of 0, and the best-performing receiving a score of 1. These normalized scores are then averaged across different datasets. For the latter, the Wilcoxon signed-ranks test (Wilcoxon, 1945) with multiple testing correction via Holm's procedure (Holm, 1979) and the Nemenyi-Friedman test (Nemenyi, 1963; Friedman, 1940) are often used. Both are nonparametric tests that check whether one model tends to rank higher than another statistically significantly. However, as we show in Finding 1 of Section 4, both methods can fail to accurately represent the statistical significance of the *extent of improvement* achieved by one model vs. another.

As such, when we compare any model pair on a given dataset, we also compute the 95% bootstrapping confidence intervals (CIs) in their *relative* test performance (e.g., $\text{AUROC}_{\text{TabPFN-v2}} - \text{AUROC}_{\text{CB}}$) by resampling the test set with replacement 1000 times. We judge a performance difference to be statistically significant if the CI does *not* overlap with 0. We use this approach to declare a win, tie, or loss for one model vs. another on each dataset, which allows us to quantify *how often one model statistically significantly outperforms the other model*.

To combine the pairwise CI-based comparisons into a coherent global ranking of models, we adapt the Elo rating system (Elo, 1978), treating each pairwise comparison on a dataset as a duel and update the ratings based on the 95% CIs in relative performance. As Elo ratings are highly sensitive to the order in which models are compared (Boubdir et al., 2024), we randomly permute the order of comparisons with 100 different random seeds, and take the average of the resulting Elo ratings.

## 4 RESULTS

**Finding 1: Comparing models based on min-max normalized evaluation metrics and ranking-based hypothesis tests can overestimate the *perceived* improvements in performance from tabular pretraining (Figure 1).** Using the original evaluation setup for TabPFN-v2 in Hollmann et al. (2025) as an example, we demonstrate the importance of accounting for statistical uncertainty in any measured pairwise performance improvement, and how it can change the conclusions about the effectiveness of large-scale tabular pretraining. We compare the performances of TabPFN-v2 against the same baselines on the same 29 classification datasets from AMLB (Gijsbers et al., 2024) (Appendix C), after optimizing the hyperparameters of each model as in Section 3. Based on *min-max normalized* test AUROC, averaged over datasets, TabPFN-v2 shows a substantial improvement over

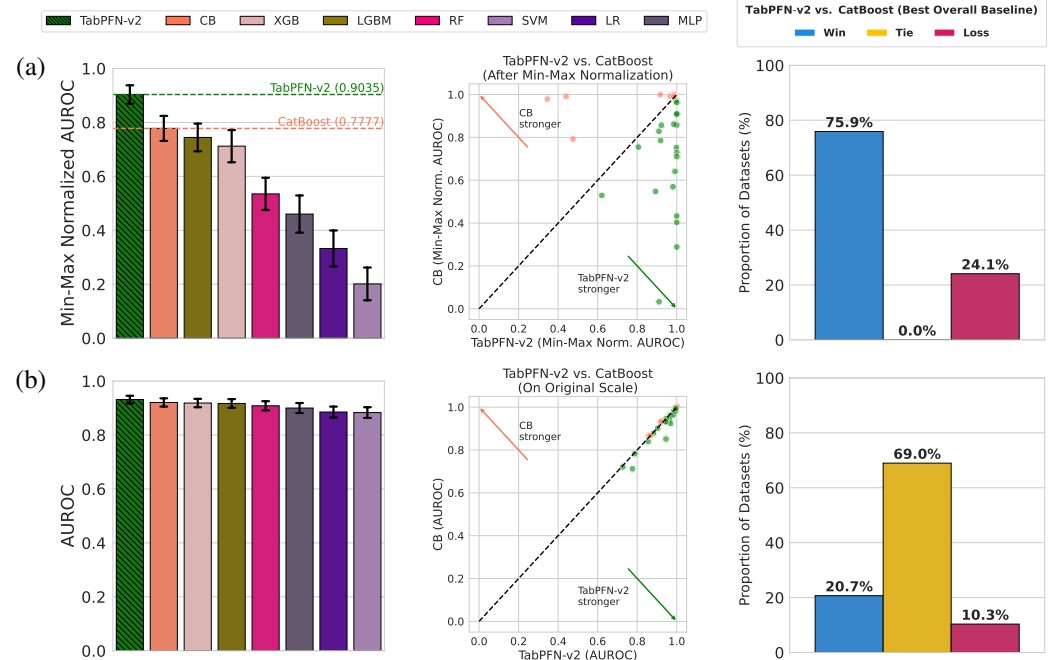

Figure 1: Comparing models based on min-max normalized evaluation metrics and ranking-based pairwise hypothesis tests can amplify the *perceived* performance improvements from tabular pre-training. To illustrate, we consider the evaluation setup for TabPFN-v2 considered in Hollmann et al. (2025), based on 29 classification tasks from AMLB. (a) *With min-max normalization*, TabPFN-v2 achieves a substantially higher average test AUROC (0.9035) than the best overall baseline (CB; 0.7777) (left), with generally large differences across datasets (middle). When we compare models based on normalized AUROC (without statistical testing), TabPFN-v2 "wins" against CB in 75.9% of datasets and "loses" in only 24.1% of datasets (right). (b) *Without normalization*, however, we see that AUROC differences *in the original scale* between TabPFN-v2 and CB are actually small across most datasets (left, middle). When we use the 95% bootstrapping CIs in relative AUROC to also test whether the *extent of improvement* on each dataset is statistically significant (Section 3), *TabPFN-v2 is virtually identical to CB performance-wise in 69% of cases*, with a slightly higher win rate (20.7%) vs. loss rate (10.3%) (right). Error bars indicate the standard error across datasets.

CB (0.9035 vs. 0.7777), which is the best overall baseline (Figure 1(a), left). Across most datasets, TabPFN-v2 shows large gains over CB in normalized AUROC, and the Wilcoxon signed-ranks test declares such a gap to be significant with a $p$-value of 0.006 ($< 0.05$) (Figure 1(a), middle).

However, when we compare the test AUROC scores for TabPFN-v2 and CB *on the original scale* (i.e., without min-max normalization), we observe that pairwise differences are small across most datasets (Figure 1(b), left & middle), and are often not statistically significant. In fact, when we compute the 95% bootstrapping CIs in relative test AUROC between TabPFN-v2 and CB on each dataset, we observe a statistical tie in 69% of cases, a statistically significant improvement of TabPFN-v2 over CB in 20.7% of cases ("win rate"), and a statistically significant underperformance of TabPFN-v2 relative to CB in 10.3% of cases ("loss rate") (Figure 4(b), right). **These results are still in favor of TabPFN-v2 over CB, but the perceived gains are far less pronounced than in the former, showing "modest" improvements.** These results suggest that, while commonly adopted, comparisons based on normalized evaluation metrics and ranking-based hypothesis tests may fail to capture *whether the pairwise performance gains themselves are significant*, which is crucial to draw reliable conclusions.

**Finding 2: In the full-data setting (i.e., using all training examples), TFMs achieve modest improvements on small classification datasets but little to no improvement on other types of datasets (Figure 2, Table 1).** Based on Finding 1, we revisit the positive conclusions of prior works on the performance benefits from large-scale tabular pretraining, using our evaluation suite introduced in Section 3. Here, we consider the full-data setting, where all of the training examples are accessible for model training or ICL. We take a closer look at two data regimes—one regime we

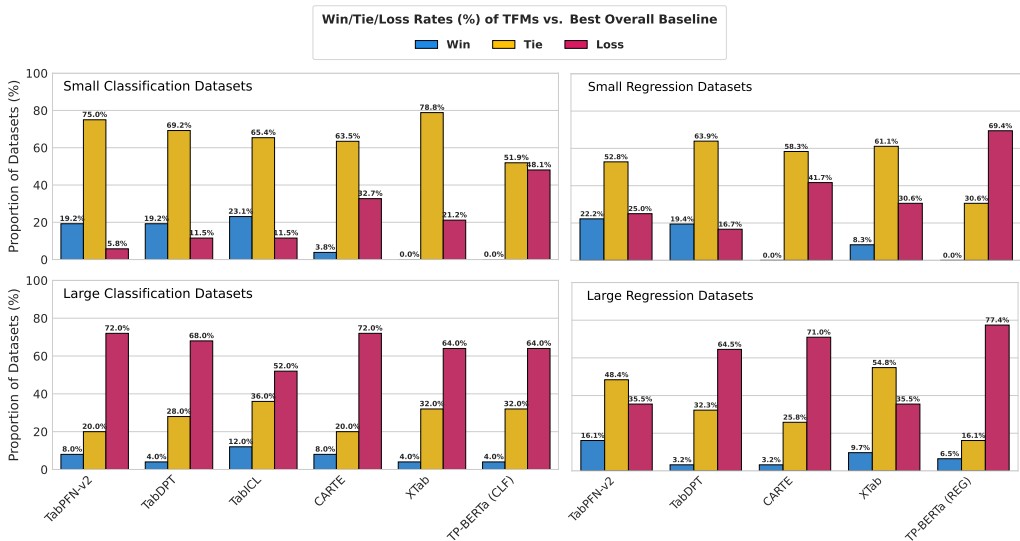

Figure 2: In the full-data setting, only 3 out of 6 TFMs improve (i.e., win more than lose) over non-pretrained baselines on small classification datasets, and no meaningful improvements are observed on large classification, small regression, or large regression datasets. We show the win, tie, and loss rates (%) of each TFM vs. the baseline with the highest Elo rating in each data regime (Figure D1). On each dataset, a TFM "wins" over the best baseline if the 95% bootstrapping confidence interval in their relative AUROC / $R^2$ lies above 0 (Section 3). *Small* datasets are those in the "TabPFN" regime with $\leq$ 10k samples, $\leq$ 500 features, and $\leq$ 10 classes, while *large* datasets refer to all others.

Table 1: The number of TFMs that win more than they lose to the best overall baseline (i.e., highest Elo rating) and their average win-loss rate gap (%) across various stratifications of evaluation datasets.

| | | Different Data Regimes | | | | | | | | | |
| --- | --- | --- | --- | --- | --- | --- | --- | --- | --- | --- | --- |
| | | Small ("TabPFN") | | Highly Categorical | | Column Semantics | | Missing Values | | Class Imbalance | |
| | | Yes | No | Yes | No | Yes | No | Yes | No | Yes | No |
| **Classification (6 TFMs)** | **# of TFMs > Best Baseline** | 3 | 0 | 0 | 2 | 0 | 0 | 2 | 1 | 0 | 0 |
| | Average (Win Rate - Loss Rate) | 9.6% | - | - | 17.3% | - | - | 12.5% | 2.8% | - | - |
| **Regression (5 TFMs)** | **# of TFMs > Best Baseline** | 1 | 0 | 0 | 0 | 1 | 0 | 0 | 0 | - | - |
| | Average (Win Rate - Loss Rate) | 2.7% | - | - | - | 12.5% | - | - | - | - | - |

refer to as the "TabPFN regime" (Hollmann et al., 2025): small datasets with $\leq$ 10k samples, $\leq$ 500 features, and $\leq$ 10 classes. The other regime consists of all other datasets, which tend to be larger with additional features and/or classes (Appendix A). We consider these settings as prior works on TFMs demonstrate that they generally show stronger performance on datasets with a small number of samples and features (McElfresh et al., 2023; Hollmann et al., 2023; 2025; Qu et al., 2025).

When directly compared against the best overall (highest Elo rating) baseline in each setting (Figure D1), only 3 out of 6 TFMs win more than they lose on small classification datasets in the "TabPFN" regime, and all TFMs show little to no improvement in the remaining settings (Figure 2). While the three tabular ICL models—TabICL, TabPFN-v2, and TabDPT—achieve the highest Elo ratings on small classification datasets (Figure D1(a)), they reach a statistical tie with the best baseline (ModernNCA) on 65–75% of datasets, with an average win-loss gap of +9.6% (Figure 2, top left). In all other settings, TFMs fail to outperform the best baselines, with both win and tie rates decreasing and the loss rates increasing substantially going from the small-data to large-data regime.

Similar trends hold under alternative stratifications of datasets by ratio of categorical features ($\geq$ 50% is considered "highly categorical"), presence of meaningful column names, missingness, and class imbalance. *In the majority of data regimes, TFMs fail to improve over the best baseline in each setting* (Table 1, Figures D2–D5). These results suggest that while large-scale tabular pretraining may improve the inductive biases of neural networks for effective tabular learning (Hollmann et al., 2023; den Breejen et al., 2025), the fundamental gains in performance may still be limited in practice.

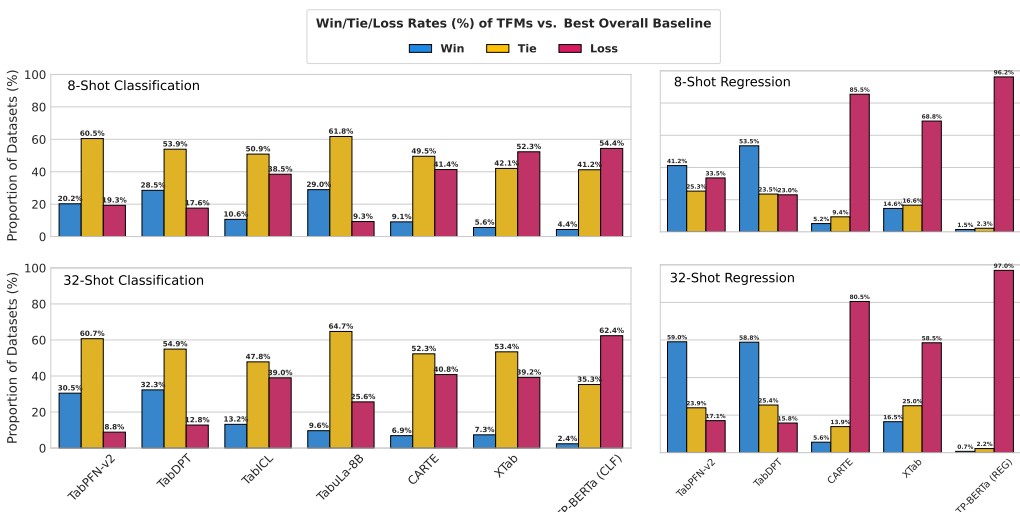

Figure 3: In the few-shot setting, only 2 out of 7 TFMs consistently improve (i.e., win more than lose) over non-pretrained baselines, and other models show little to no improvement at all. We show the pairwise win, tie, and loss rates (%) of each TFM vs. the baseline with the highest Elo rating in each $k$-shot setting (Figure D6). TabPFN-v2 and TabDPT scale better on regression tasks, although with sufficient examples, the performance gains over non-pretrained baselines become small (Figure 2).

**Finding 3: In the few-shot setting, only a few TFMs exhibit better sample efficiency compared to non-pretrained baselines, and the majority of models show little to no efficiency gains (Figure 3).** To investigate whether large-scale tabular pretraining leads to consistent improvements in sample efficiency, we evaluate all TFMs in the few-shot setting, as described in Section 3. As in Finding 2, we measure the win, tie, loss rates (%) of each TFM vs. the best overall baseline (highest Elo rating) in each $k$-shot setting, based on the 95% bootstrapping CIs in relative AUROC or $R^2$. We average all of the results over five random seeds that control the subsampling of training examples in each $k$-shot setting, in order to obtain a stable measure of few-shot performance.

On both classification and regression tasks in the 8-shot and 32-shot settings, only 2 out of 7 TFMs show consistent improvements over the best overall baseline (CB; Figure D6), with TabPFN-v2 and TabDPT being the only models that win more than they lose across all cases (Figure 3). TabuLa-8B significantly outperforms CB in the 8-shot setting but underperforms it in the 32-shot setting ($+19.7\%$ $\rightarrow -16\%$), showing limited sample efficiency despite being pretrained on 4M tables (Gardner et al., 2024). Other models such as CARTE, TP-BERTa, and XTab substantially underperform CB, with loss rates going up to 97%. These results suggest that large-scale tabular pretraining can lead to improved sample efficiency, but the benefits remain inconsistent across different models and prediction tasks.

**Finding 4: Head-to-head comparisons of three TFMs vs. their non-pretrained counterparts show that tabular pretraining (i) improves LLMs' tabular ICL capabilities, (ii) but offers no clear performance benefits for other models, yielding limited overall gains (Figure 4).** To fully isolate the impact of tabular pretraining on downstream predictive performance, we compare three TFMs (TabuLa-8B, TP-BERTa, and XTab) directly against their non-pretrained counterparts. For each comparison, the only difference between models lies in tabular pretraining (i.e., each comparison uses the same model architecture and size). For TabuLa-8B, we compare against Llama-3-8B in the few-shot classification setting (up to 32 examples per class due to context window limits), while optimizing the prompt for Llama-3-8B on each dataset and few-shot setting (Section 3). For TP-BERTa, we compare it against TP-BERTa (RoBERTa) and TP-BERTa (Random), which correspond to initializing the model with RoBERTa and random weights, respectively. For XTab, we compare it against XTab (Random)—its randomly initialized counterpart. We exclude the tabular ICL models, as their non-pretrained counterparts (random weights) would be incapable of ICL.

In the zero-shot setting, TabuLa-8B outperforms Llama-3-8B by a noticeable margin, achieving win and loss rates of 29.6% and 13%, while reaching a tie in the remaining 57.4% of datasets (Figure 4(a)). As the number of few-shot examples increases, TabuLa-8B outperforms Llama-3-8B in a much larger fraction of cases, modulo the statistical ties that remain at around 50%. This is consistent with the

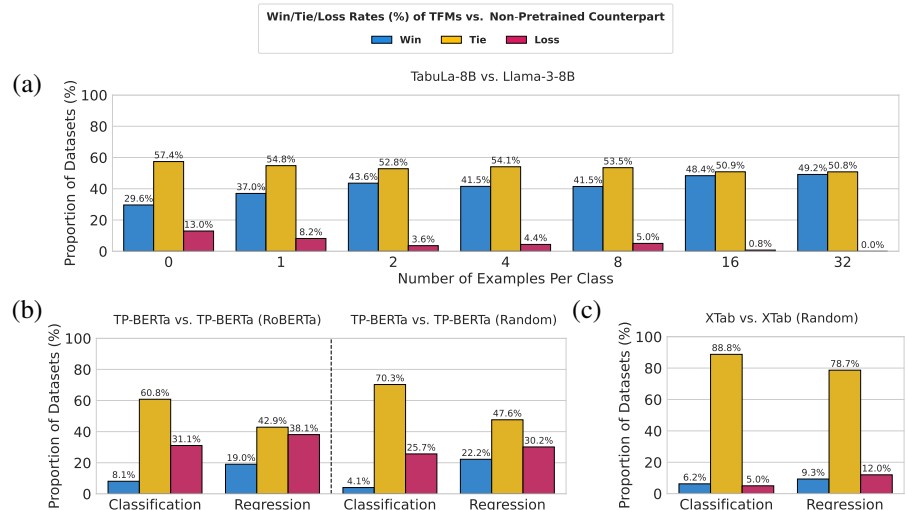

Figure 4: In an apples-to-apples comparison between three TFMs vs. their non-pretrained counterparts, TabuLa-8B shows statistically significant improvements, while TP-BERTa and XTab do not. (a) Win, tie, loss rates (%) for TabuLa-8B vs. Llama-3-8B in few-shot classification settings. The two models reach a tie on 40–60% of datasets, but TabuLa-8B consistently wins more than it loses to Llama-3-8B. (b) Win, tie, loss rates (%) for TP-BERTa vs. TP-BERTa (RoBERTa) & TP-BERTa (Random) on both classification and regression datasets in the full-data setting. (c) Win, tie, loss rates (%) for XTab vs. XTab (Random) on both classification and regression datasets in the full-data setting.

findings of Gardner et al. (2024), and suggests that (i) fine-tuning an LLM for tabular classification tasks can improve the stability of tabular ICL; and (ii) it is challenging to elicit the same scaling behavior from an LLM only trained on text just by optimizing the details of the prompt. Meanwhile, TabuLa-8B still fails to consistently outperform the non-pretrained baselines (Figure 3(a)), which undermines the practical utility of adapting an LLM for tabular prediction.

On the other hand, both TP-BERTa and XTab fail to improve over their non-pretrained counterparts, on both classification and regression tasks in the full-data setting. TP-BERTa generally performs worse than TP-BERTa (RoBERTa) and TP-BERTa (Random), with a higher loss rate than win rate (Figure 4(b)). XTab is virtually indistinguishable from XTab (Random) performance-wise, reaching a tie in 80–90% of cases and achieving almost identical win and loss rates (Figure 4(c)). These results indicate that neither model benefits from tabular pretraining.

## 5 Discussion and Conclusion

In this work, we conducted a large-scale evaluation of seven open-source TFMs across 88 classification and 82 regression datasets, testing whether the observed performance differences are statistically significant and showing that neglecting this may lead to overly optimistic conclusions about the performance benefits from large-scale tabular pretraining. Our results revealed that tabular pretraining yields modest gains in narrow settings—such as improved tabular ICL capabilities for LLMs (e.g., TabuLa-8B) and better sample efficiency for models like TabPFN-v2 and TabDPT—but these benefits are inconsistent, often tied, and generally insufficient to surpass strong non-pretrained baselines (e.g., GBDTs). In the full-data regime, improvements are largely confined to small classification datasets, with little to no meaningful advantage elsewhere. In few-shot settings, only 2 out of 7 TFMs reliably outperformed baselines, while others underperform or show no benefit from pretraining. These findings suggest that current approaches to large-scale tabular pretraining may offer limited performance benefits, showing room for improvement in methods for effective tabular transfer learning.

**Limitations.** Due to space constraints, we include our discussion of limitations in Appendix E.

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

## A  ADDITIONAL DETAILS ON DATASETS

**Dataset Summary Statistics.**    In Tables A1 and A2, we list all OpenML classification and regression datasets selected for evaluation from the candidate pool constructed from existing benchmarks ("Datasets" in Section 3). We provide all of the OpenML task IDs that can be used to load the exact same dataset and dataset splits as used in our experiments. As discussed in Section 3, the datasets in our benchmark cover a wide range of settings, in terms of sample size, number of features, ratio of categorical/numerical features, presence of meaningful column names, missingness, etc.

Table A1: List of all OpenML classification datasets used in our evaluation suite. "Class Imbalance" denotes the ratio of the # of samples in the minority class to that in the majority class. "Cat. Ratio" denotes the proportion of features that are categorical.

| OpenML Task ID | # Samples | # Features | # Numerical Features | # Categorical Features | Meaningful Col. Names | # Classes | Class Imbalance | Cat. Ratio | Missing Values |
|---|---|---|---|---|---|---|---|---|---|
| 23 | 1473 | 9 | 2 | 7 | Yes | 3 | 0.529 | 0.778 | No |
| 25 | 368 | 26 | 7 | 19 | Yes | 2 | 0.586 | 0.731 | Yes |
| 29 | 690 | 15 | 6 | 9 | No | 2 | 0.802 | 0.6 | Yes |
| 31 | 1000 | 20 | 7 | 13 | Yes | 2 | 0.429 | 0.65 | No |
| 37 | 768 | 8 | 8 | 0 | Yes | 2 | 0.536 | 0.0 | No |
| 50 | 294 | 13 | 6 | 7 | Yes | 2 | 0.564 | 0.538 | Yes |
| 206 | 39366 | 9 | 0 | 9 | Yes | 2 | 0.532 | 1.0 | No |
| 219 | 45312 | 8 | 7 | 1 | Yes | 2 | 0.738 | 0.125 | No |
| 2075 | 4177 | 8 | 7 | 1 | Yes | 28 | 0.001 | 0.125 | No |
| 2079 | 736 | 19 | 14 | 5 | Yes | 5 | 0.491 | 0.263 | Yes |
| 2146 | 55296 | 9 | 2 | 7 | Yes | 3 | 0.528 | 0.778 | No |
| 3560 | 797 | 4 | 0 | 4 | Yes | 6 | 0.794 | 1.0 | No |
| 3561 | 672 | 9 | 5 | 4 | Yes | 2 | 0.5 | 0.444 | No |
| 3899 | 15545 | 5 | 5 | 0 | Yes | 2 | 0.489 | 0.0 | No |
| 3903 | 1563 | 37 | 37 | 0 | Yes | 2 | 0.114 | 0.0 | No |
| 3904 | 10885 | 21 | 21 | 0 | No | 2 | 0.24 | 0.0 | Yes |
| 3913 | 522 | 21 | 21 | 0 | No | 2 | 0.258 | 0.0 | No |
| 3917 | 2109 | 21 | 21 | 0 | No | 2 | 0.183 | 0.0 | No |
| 7592 | 48842 | 14 | 6 | 8 | Yes | 2 | 0.315 | 0.571 | No |
| 9906 | 1100 | 12 | 8 | 4 | No | 5 | 0.502 | 0.333 | No |
| 9908 | 2500 | 100 | 58 | 42 | No | 3 | 0.167 | 0.42 | No |
| 9910 | 3751 | 1776 | 1776 | 0 | No | 2 | 0.844 | 0.0 | No |
| 9952 | 5404 | 5 | 5 | 0 | No | 2 | 0.415 | 0.0 | No |
| 9957 | 1055 | 41 | 41 | 0 | No | 2 | 0.509 | 0.0 | No |
| 9959 | 7400 | 20 | 20 | 0 | No | 2 | 0.981 | 0.0 | No |
| 9971 | 583 | 10 | 9 | 1 | No | 2 | 0.401 | 0.1 | No |
| 9976 | 2600 | 500 | 500 | 0 | No | 2 | 1.0 | 0.0 | No |
| 9985 | 6118 | 51 | 51 | 0 | No | 6 | 0.19 | 0.0 | No |
| 10101 | 748 | 4 | 4 | 0 | No | 2 | 0.312 | 0.0 | No |
| 14954 | 540 | 35 | 18 | 17 | Yes | 2 | 0.731 | 0.486 | Yes |
| 14969 | 9873 | 32 | 32 | 0 | No | 5 | 0.338 | 0.0 | No |
| 125920 | 500 | 12 | 1 | 11 | No | 2 | 0.724 | 0.917 | No |
| 145941 | 2800 | 26 | 6 | 20 | No | 5 | 0.019 | 0.769 | No |
| 146065 | 601 | 6 | 0 | 6 | No | 2 | 0.522 | 1.0 | No |
| 146177 | 1600 | 20 | 0 | 20 | No | 2 | 1.0 | 1.0 | No |
| 146195 | 67557 | 42 | 0 | 42 | No | 3 | 0.145 | 1.0 | No |
| 146205 | 3200 | 7 | 0 | 7 | No | 10 | 0.792 | 1.0 | No |
| 146217 | 1599 | 11 | 11 | 0 | Yes | 6 | 0.015 | 0.0 | No |
| 167119 | 44819 | 6 | 6 | 0 | Yes | 3 | 0.188 | 0.0 | No |
| 167120 | 96320 | 21 | 21 | 0 | No | 2 | 0.98 | 0.0 | No |
| 167141 | 5000 | 20 | 16 | 4 | Yes | 2 | 0.165 | 0.2 | No |
| 189773 | 20000 | 20 | 20 | 0 | No | 5 | 0.067 | 0.0 | No |
| 189922 | 3153 | 970 | 970 | 0 | No | 2 | 0.967 | 0.0 | No |
| 189939 | 1294 | 140 | 139 | 1 | Yes | 11 | 0.081 | 0.007 | Yes |
| 190412 | 100 | 9920 | 9920 | 0 | No | 2 | 0.786 | 0.0 | No |
| 359974 | 4898 | 11 | 11 | 0 | No | 7 | 0.002 | 0.0 | No |
| 360676 | 43825 | 9 | 9 | 0 | Yes | 2 | 0.942 | 0.0 | No |
| 360679 | 1834 | 9 | 9 | 0 | Yes | 2 | 0.97 | 0.0 | No |
| 360721 | 43825 | 9 | 9 | 0 | Yes | 2 | 0.943 | 0.0 | No |
| 360791 | 43825 | 9 | 9 | 0 | Yes | 2 | 0.931 | 0.0 | No |
| 360797 | 1834 | 5 | 5 | 0 | Yes | 2 | 0.993 | 0.0 | No |
| 360801 | 43825 | 9 | 9 | 0 | Yes | 2 | 0.888 | 0.0 | No |
| 360822 | 43825 | 10 | 10 | 0 | Yes | 2 | 0.943 | 0.0 | No |
| 360832 | 1833 | 9 | 9 | 0 | Yes | 2 | 0.973 | 0.0 | No |
| 360839 | 1832 | 5 | 5 | 0 | Yes | 2 | 0.947 | 0.0 | No |
| 361056 | 20634 | 8 | 8 | 0 | Yes | 2 | 1.0 | 0.0 | No |
| 361057 | 2554 | 11 | 11 | 0 | Yes | 2 | 1.0 | 0.0 | No |
| 361060 | 38474 | 7 | 7 | 0 | Yes | 2 | 1.0 | 0.0 | No |
| 361063 | 13488 | 16 | 16 | 0 | No | 2 | 1.0 | 0.0 | No |
| 361064 | 5188 | 20 | 20 | 0 | Yes | 2 | 1.0 | 0.0 | No |
| 361065 | 13376 | 9 | 9 | 0 | No | 2 | 1.0 | 0.0 | No |
| 361066 | 10578 | 7 | 7 | 0 | No | 2 | 1.0 | 0.0 | No |
| 361069 | 940160 | 24 | 24 | 0 | No | 2 | 1.0 | 0.0 | No |
| 361070 | 7608 | 20 | 20 | 0 | No | 2 | 1.0 | 0.0 | No |
| 361111 | 7608 | 23 | 20 | 3 | No | 2 | 1.0 | 0.13 | No |
| 361112 | 5032 | 45 | 34 | 11 | No | 2 | 1.0 | 0.244 | No |
| 361114 | 4970 | 11 | 5 | 6 | No | 2 | 1.0 | 0.545 | No |
| 361115 | 111762 | 32 | 29 | 3 | Yes | 2 | 1.0 | 0.094 | No |
| 361116 | 16644 | 16 | 8 | 8 | Yes | 2 | 1.0 | 0.5 | No |
| 361278 | 10000 | 22 | 22 | 0 | Yes | 2 | 1.0 | 0.0 | No |

*Continued on next page*

| Task ID | # Samples | # Features | # Numerical Features | # Categorical Features | Meaningful Col. Names | # Classes | Class Imbalance | Cat. Ratio | Missing Values |
|---|---|---|---|---|---|---|---|---|---|
| 361302 | 12330 | 17 | 5 | 12 | Yes | 2 | 0.183 | 0.706 | No |
| 361304 | 10000 | 10 | 4 | 6 | Yes | 2 | 0.256 | 0.6 | No |
| 361306 | 23548 | 10 | 3 | 7 | Yes | 2 | 0.319 | 0.7 | No |
| 361309 | 26677 | 13 | 2 | 11 | Yes | 3 | 0.146 | 0.846 | No |
| 361315 | 10999 | 9 | 5 | 4 | Yes | 2 | 0.676 | 0.444 | No |
| 361316 | 5032 | 45 | 34 | 11 | No | 2 | 1.0 | 0.244 | No |
| 361930 | 1649 | 104 | 9 | 95 | No | 2 | 0.185 | 0.913 | No |
| 362772 | 2000 | 7 | 3 | 4 | Yes | 2 | 0.803 | 0.571 | No |
| 363088 | 9871 | 23 | 21 | 2 | Yes | 2 | 0.922 | 0.087 | No |
| 363322 | 20640 | 8 | 8 | 0 | Yes | 2 | 0.999 | 0.0 | No |
| 363323 | 70000 | 11 | 5 | 6 | Yes | 2 | 0.999 | 0.545 | No |
| 363328 | 2400 | 30 | 30 | 0 | No | 2 | 1.0 | 0.0 | No |
| 363329 | 1190 | 11 | 11 | 0 | Yes | 2 | 0.892 | 0.0 | No |
| 363331 | 1723 | 13 | 9 | 4 | Yes | 2 | 0.128 | 0.308 | No |
| 363334 | 1340 | 20 | 19 | 1 | Yes | 2 | 0.613 | 0.05 | Yes |
| 363337 | 1188 | 22 | 22 | 0 | No | 2 | 1.0 | 0.0 | No |
| 363338 | 1243 | 21 | 21 | 0 | No | 2 | 0.313 | 0.0 | No |
| 363339 | 7043 | 19 | 4 | 15 | Yes | 2 | 0.361 | 0.789 | Yes |

Table A2: List of all OpenML regression datasets used in our evaluation suite. "Cat. Ratio" denotes the proportion of features that are categorical.

| OpenML Task ID | # Samples | # Features | # Numerical Features | # Categorical Features | Meaningful Col. Names | Cat. Ratio | Missing Values |
|---|---|---|---|---|---|---|---|
| 2289 | 9517 | 6 | 6 | 0 | Yes | 0.0 | No |
| 2306 | 40768 | 10 | 10 | 0 | No | 0.0 | No |
| 2309 | 22784 | 8 | 8 | 0 | No | 0.0 | No |
| 2313 | 8192 | 8 | 8 | 0 | No | 0.0 | No |
| 4708 | 6435 | 36 | 36 | 0 | No | 0.0 | No |
| 4772 | 8192 | 32 | 32 | 0 | No | 0.0 | No |
| 4831 | 6574 | 14 | 14 | 0 | No | 0.0 | No |
| 4881 | 8192 | 32 | 32 | 0 | No | 0.0 | No |
| 4885 | 40768 | 10 | 10 | 0 | No | 0.0 | No |
| 4891 | 8192 | 8 | 8 | 0 | No | 0.0 | No |
| 5012 | 526 | 5 | 3 | 2 | No | 0.4 | No |
| 7320 | 31104 | 9 | 2 | 7 | No | 0.778 | No |
| 7323 | 17496 | 9 | 6 | 3 | Yes | 0.333 | No |
| 7393 | 10886 | 10 | 4 | 6 | Yes | 0.6 | No |
| 189931 | 2108 | 25 | 22 | 3 | Yes | 0.12 | No |
| 190418 | 10738 | 14 | 14 | 0 | Yes | 0.0 | No |
| 233169 | 6277 | 6 | 5 | 1 | No | 0.167 | No |
| 359930 | 2178 | 3 | 3 | 0 | No | 0.0 | No |
| 359931 | 576 | 11 | 0 | 11 | Yes | 1.0 | No |
| 359936 | 16599 | 18 | 18 | 0 | No | 0.0 | No |
| 359939 | 8885 | 261 | 261 | 0 | No | 0.0 | No |
| 359940 | 8885 | 212 | 212 | 0 | No | 0.0 | No |
| 359943 | 581835 | 18 | 9 | 9 | Yes | 0.5 | No |
| 359949 | 21613 | 21 | 20 | 1 | Yes | 0.048 | No |
| 360879 | 45918 | 17 | 17 | 0 | No | 0.0 | No |
| 361074 | 16599 | 16 | 16 | 0 | No | 0.0 | No |
| 361075 | 7797 | 613 | 613 | 0 | No | 0.0 | No |
| 361078 | 20640 | 8 | 8 | 0 | Yes | 0.0 | No |
| 361079 | 22784 | 16 | 16 | 0 | No | 0.0 | No |
| 361080 | 53940 | 6 | 6 | 0 | Yes | 0.0 | No |
| 361082 | 17379 | 6 | 6 | 0 | Yes | 0.0 | No |
| 361083 | 581835 | 9 | 9 | 0 | Yes | 0.0 | No |
| 361084 | 21613 | 15 | 15 | 0 | Yes | 0.0 | No |
| 361087 | 13932 | 13 | 13 | 0 | Yes | 0.0 | No |
| 361088 | 21263 | 79 | 79 | 0 | Yes | 0.0 | No |
| 361089 | 20640 | 8 | 8 | 0 | Yes | 0.0 | No |
| 361090 | 18063 | 5 | 5 | 0 | Yes | 0.0 | No |
| 361091 | 515345 | 90 | 90 | 0 | No | 0.0 | No |
| 361092 | 8885 | 62 | 42 | 20 | No | 0.323 | No |
| 361095 | 166821 | 9 | 4 | 5 | Yes | 0.556 | No |
| 361098 | 10692 | 11 | 8 | 3 | Yes | 0.273 | No |
| 361100 | 39644 | 59 | 45 | 14 | Yes | 0.237 | No |
| 361102 | 21613 | 17 | 15 | 2 | Yes | 0.118 | No |
| 361234 | 4177 | 8 | 7 | 1 | Yes | 0.125 | No |
| 361235 | 1503 | 5 | 5 | 0 | Yes | 0.0 | No |
| 361236 | 2043 | 7 | 5 | 2 | No | 0.286 | No |
| 361237 | 1030 | 8 | 8 | 0 | Yes | 0.0 | No |

*Continued on next page*

| Task ID | # Samples | # Features | # Numerical Features | # Categorical Features | Meaningful Col. Names | Cat. Ratio | Missing Values |
|---|---|---|---|---|---|---|---|
| 361241 | 45730 | 9 | 9 | 0 | No | 0.0 | No |
| 361242 | 21263 | 81 | 81 | 0 | Yes | 0.0 | No |
| 361243 | 1059 | 116 | 116 | 0 | No | 0.0 | No |
| 361244 | 1066 | 9 | 1 | 8 | Yes | 0.889 | No |
| 361249 | 4898 | 11 | 11 | 0 | Yes | 0.0 | No |
| 361250 | 1599 | 11 | 11 | 0 | Yes | 0.0 | No |
| 361251 | 10000 | 12 | 12 | 0 | No | 0.0 | No |
| 361252 | 68784 | 18 | 16 | 2 | Yes | 0.111 | No |
| 361255 | 20640 | 8 | 8 | 0 | Yes | 0.0 | No |
| 361258 | 8192 | 8 | 8 | 0 | No | 0.0 | No |
| 361259 | 8192 | 32 | 32 | 0 | No | 0.0 | No |
| 361260 | 13932 | 15 | 15 | 0 | Yes | 0.0 | No |
| 361261 | 28155 | 6 | 2 | 4 | Yes | 0.667 | No |
| 361264 | 1156 | 5 | 1 | 4 | Yes | 0.8 | No |
| 361266 | 21613 | 21 | 17 | 4 | Yes | 0.19 | No |
| 361267 | 10692 | 9 | 5 | 4 | Yes | 0.444 | No |
| 361269 | 22272 | 11 | 4 | 7 | Yes | 0.636 | No |
| 361272 | 19178 | 28 | 27 | 1 | Yes | 0.036 | No |
| 361616 | 1232 | 14 | 8 | 6 | Yes | 0.429 | Yes |
| 361618 | 517 | 12 | 10 | 2 | Yes | 0.167 | No |
| 361619 | 649 | 30 | 13 | 17 | Yes | 0.567 | No |
| 361621 | 908 | 6 | 6 | 0 | No | 0.0 | No |
| 361622 | 804 | 17 | 17 | 0 | Yes | 0.0 | No |
| 361623 | 3107 | 6 | 6 | 0 | Yes | 0.0 | No |
| 362387 | 1038 | 12 | 11 | 1 | Yes | 0.083 | Yes |
| 362390 | 1234 | 8 | 5 | 3 | Yes | 0.375 | No |
| 362394 | 1538 | 7 | 6 | 1 | Yes | 0.143 | No |
| 362418 | 18249 | 13 | 11 | 2 | Yes | 0.154 | No |
| 362589 | 14480 | 29 | 28 | 1 | Yes | 0.034 | No |
| 363138 | 18182 | 12 | 8 | 4 | Yes | 0.333 | No |
| 363343 | 2394 | 7 | 7 | 0 | No | 0.0 | No |
| 363344 | 78732 | 10 | 7 | 3 | No | 0.3 | No |
| 363345 | 59049 | 9 | 9 | 0 | No | 0.0 | No |
| 363346 | 28155 | 6 | 2 | 4 | Yes | 0.667 | No |
| 363347 | 21613 | 19 | 18 | 1 | Yes | 0.053 | No |

# B  ADDITIONAL DETAILS ON MODEL TRAINING AND HYPERPARAMETER OPTIMIZATION

Here, we provide additional details on how we train each model and optimize the hyperparameters.

## B.1  PROMPT OPTIMIZATION FOR FEW-SHOT CLASSIFICATION WITH LLAMA-3-8B

Following Jeong et al. (2024), we first (i) construct a context-free grammar of plausible prompt formats for tabular classification with LLMs; and then (ii) select the best combination of prompt format, task instruction, and "serialization" method (i.e., format used for representing tabular features as text) that results in the highest validation AUROC to use for final evaluation. We clarify that one important difference with Jeong et al. (2024) is that we do not sample the few-shot examples to use for ICL, as they are assumed to be given (i.e., they are the only examples accessible by the model).

**Context-free grammar of prompt formats.** Using the Backus-Naur notation as in Jeong et al. (2024), let $H_i$ be the instruction header (e.g., "### Instruction:"), $H_f$ be the feature header (e.g., "### Features:"), and $H_a$ be the answer header (e.g., "### Answer:"). We define each of these three headers as follows:

$$H_i(f_{\text{case}}, d_q, s) ::= f_{\text{case}}(d_i)s\langle\text{text}\rangle,$$
$$H_f(f_{\text{case}}, d_f, s) ::= f_{\text{case}}(d_f)s\langle\text{text}\rangle,$$
$$H_a(f_{\text{case}}, d_a, s) ::= f_{\text{case}}(d_a)s\langle\text{text}\rangle,$$

where $f_{\text{case}} \in \mathcal{F}_{\text{case}}$ denotes the casing function (e.g., $x \mapsto$ "### " + x, $x \mapsto$ x.upper()), $d_i \in D_i$ denotes the instruction descriptor (e.g., "Instruction"), $d_f \in D_f$ denotes the feature descriptor (e.g., "Features"), $d_a \in D_a$ denotes the answer descriptor (e.g., "Answer"), $s \in S$ denotes the header separator (e.g., ':'), and $\langle\text{text}\rangle$ denotes a text placeholder. Notably, the text placeholder in $H_f$ demarcates where the serialized version of the tabular rows (e.g., "height is 183; systolic

`blood pressure is 79...")` is placed. The full prompt format $P(f_{\text{case}}, d_i, d_f, d_a, s)$ is then constructed by concatenating all of the headers, while adding space $t \in T$ (e.g., "`\n`") in-between:

$$P ::= H_i t H_f t H_a, \tag{1}$$

where the arguments for each header are excluded for notational simplicity. We instantiate the above grammar with the descriptors, separators, and functions below.

**- Descriptors:**

$$D_i = \{\text{"Instruction"}, \text{"Task"}, \text{""}\},$$
$$D_f = \{\text{"Description"}, \text{"Input Table"}, \text{"Features"}\};$$
$$D_a = \{\text{"Answer"}, \text{"Output"}; \text{"Response"}, \text{"Prediction"}\}.$$

**- Separators:**

$S = \{\text{": "}, \text{" : "}, \text{" :: "}, \text{":\textbackslash n"}, \text{"= "}, \text{" = "}, \text{" == "}, \text{"=\textbackslash n"}, \text{" - "},$
$\quad\text{" -- "}, \text{"---"}, \text{"\textbackslash n"}, \text{"\textbackslash n\textbackslash n"}\}.$

**- Spaces:**

$$T = \{\text{"\textbackslash n"}, \text{"\textbackslash\textbackslash"}, \text{" || "}, \text{" "}\}.$$

**- Casing Functions:**

$\mathcal{F}_{\text{case}} = \{\text{x} \mapsto \text{x}, \text{x} \mapsto \text{x.title()}, \text{x} \mapsto \text{x.upper()}, \text{x} \mapsto \text{x.lower()}, \text{x} \mapsto \text{"\#\#\# "} + \text{x}\}.$

To randomly sample a prompt format accepted by the grammar, we randomly sample each of these components and construct the full prompt format, following Equation equation 1.

**Task instruction.** We sample the task instruction to provide to the model from the following list of predefined instructions, where $\langle \rangle$ is a placeholder for the target column name:

- Based on the provided column values of a row from a table, predict the value of the $\langle \rangle$ column.
- Using the given column values from a row in a table, determine the value for the $\langle \rangle$ column.
- Given the values of other columns in a table row, predict the value for the $\langle \rangle$ column.
- From the provided table row data, infer the value of the $\langle \rangle$ column.
- Based on the information in the row's column values, estimate the value of the $\langle \rangle$ column.
- Using the row's column values from the table, calculate the value of the $\langle \rangle$ column.
- With the available column values for a table row, predict what the $\langle \rangle$ column contains.
- Taking into account the values from the other columns in the row, determine the $\langle \rangle$ column's value.
- Analyze the row's column values to forecast the value of the $\langle \rangle$ column.
- Given the data for a row's columns, identify the value of the $\langle \rangle$ column.
- Predict the value of the $\langle \rangle$ column using the values of other columns in the row from the table.
- Identify the value of the $\langle \rangle$ column based on the data provided for the other columns in the row.
- Using the row's data from a table, forecast the value of the $\langle \rangle$ column.
- Determine the $\langle \rangle$ column's value by analyzing the row's other column values from the table.
- Relying on the provided row data, calculate the value of the $\langle \rangle$ column.
- From the given table row values, deduce what the $\langle \rangle$ column contains.
- Based on the row's column values, ascertain the value of the $\langle \rangle$ column in the table.
- Leverage the provided row data to infer the value of the $\langle \rangle$ column from the table.
- Examine the column values in a table row to predict the entry for the $\langle \rangle$ column.
- Given the data for the row in the table, determine what the $\langle \rangle$ column should be.

The task instruction is then followed by the phrase "Possible choices are: $\langle \text{labels} \rangle$", where $\langle \text{labels} \rangle$ is a placeholder for all of the possible class labels.

**Serialization method.** We consider a total of 7 different serialization methods, which are reported to work well in prior works (Dinh et al., 2022; Hegselmann et al., 2023; Wang et al., 2024; Fang et al., 2024). For illustration, suppose that we are serializing a table with two features/columns "height" and "weight", which each take values 180 and 80. Below, we show the outcome of serializing a row in this table according to each serialization method:

- *List*: "- height: 180\- weight: 80\n"
- *Sentence*: "The height is 180. The weight is 80."
- *Attribute-Value*: "{height: 180; weight: 80.}"
- *Equation*: "height=180, weight=80."
- *Markdown*: "| height | weight |\n| — | — |\| 180 | 80 |"
- *CSV*: "height,weight\n180,80"
- *HTML*: "<table>\n\t<tr>\n\t \t<th>height</th>\n\t\t<th>weight</th>\n\t</tr>\n\t<tr>\n\t \t<td>180</td>\n\t \t<td>80</td>\n\t</tr>\n</table>"

Meanwhile, we round all of the floating-point values to the fourth decimal place by default, in order to avoid excessively long tokenization for tables with many numerical features.

**Sampling and selection of prompts.** For each $k$-shot setting and run (i.e., one of the 5 repeated trials with different random seeds), we further split given examples into a training and validation set, in order to select the prompt (i.e., combination of prompt format, task instruction, and serialization) that results in the best validation performance. Note that if $k \in \{0, 1\}$, it is not feasible to split into a training and validation set, so we only perform this procedure for $k \geq 2$. After obtaining the train and validation sets, we randomly sample 30 different combinations of the prompt format, task instruction, and serialization method described above. Below, we show examples of 2-shot prompts sampled based according to the procedure above (classification dataset with task ID 31, with only 3 features shown). Using the examples in the training set for ICL, we measure the validation AUROC for all of the randomly sampled prompts, and use the best prompt for final evaluation on the test set. This procedure is performed to account for the sensitivity of LLMs to the choice of prompts (Sclar et al., 2024; Jeong et al., 2024), which is critical for an apples-to-apples comparison of the best performances. Meanwhile, as the predefined prompt search space does not cover all possible prompts, the performance measured under this framework may still underestimate the "best" performance achievable with an LLM.

---

**Example 1**

Based on the provided column values of a row from a table, predict the value of the "class" column. Possible choices are ['bad', 'good'].

Description : - checking_status: 0<=X<200
- duration: 48.0
- credit_amount: 14421.0

Response: bad

Description : - checking_status: 0<=X<200
- duration: 18.0
- credit_amount: 1056.0

Response: good

Description : - checking_status: no checking
- duration: 42.0
- credit_amount: 3161.0

Response:

---

Table B1: Hyperparameter search space for TP-BERTa. $\lambda$ is the coefficient applied to the special regularization term for relative magnitude tokenization (see Equation (11) of Yan et al. (2024)).

| Hyperparameter | Distribution |
|---|---|
| Learning Rate | LogUniform(1e-6,5e-5) |
| Weight Decay | LogUniform(1e-6,0.5) |
| $\lambda$ | LogUniform(1e-6,0.2) |

Table B2: Hyperparameter search space for CARTE.

| Hyperparameter | Distribution |
|---|---|
| Learning Rate | LogUniform(1e-4,1e-2) |
| Dropout | LogUniform(1e-6,0.5) |
| Freeze Pretrained Weights | Categorical([True, False]) |

Table B3: Hyperparameter search space for XTab.

| Hyperparameter | Distribution |
|---|---|
| Learning Rate | LogUniform(1e-5,1e-3) |
| Weight Decay | LogUniform(1e-5,1e-3) |

**Example 2**

Analyze the row's column values to forecast the value of the "class" column. ‖ ### Input Table == checking_status=0<=X<200, duration=48.0, credit_amount=14421.0 ‖ ### Output == bad ‖ ### Input Table == checking_status=0<=X<200, duration=18.0, credit_amount=1056.0 ‖ ### Output == good ‖ ### Input Table == checking_status=0<=X<200, duration=42.0, credit_amount=3161.0 ‖ ### Output ==

### B.2 HYPERPARAMETER OPTIMIZATION

Here, we detail all of the hyperparameter search spaces for all of the TFMs (if training is involved) and all of the baselines. As discussed in Section 3, we optimize the hyperparameters of all models by randomly sampling 100 configurations (except TP-BERTa, for which we sample 30 configurations) based on the Tree-structured Parzen Estimator (TPE; Watanabe, 2023) and selecting the configuration that results in the best validation performance (AUROC ($\uparrow$) for classification, RMSE ($\downarrow$) for regression). Meanwhile, for each model that undergoes mini-batch training, we use the largest batch size affordable by the model on each given dataset, based on a predefined list of batch sizes, in order to better handle out-of-memory errors and avoid excessively long runtimes (especially since we do not impose arbitrary time limits during hyperparameter optimization). If none of the batch sizes work for a model on some dataset, we exclude this (model, dataset) pair from evaluation. For TP-BERTa, we try the batch sizes [64,32,16,8,4,2,1] in order. For CARTE, we try the batch sizes [256,128,64,32,16,8,4,2,1] in order. For all of the deep baselines, we try the batch sizes [512,256,128,64,32,16,8,4,2,1] in order. In Tables B1–B18, we detail the hyperparameter search space used for each model.

### C REPLICATION OF TABPFN-V2 EXPERIMENTS

Here, we provide additional details on our process for replicating the classification setup considered in Hollmann et al. (2025), where TabPFN-v2 is compared against several baselines—CatBoost (CB; Prokhorenkova et al., 2018), XGBoost (XGB; Chen & Guestrin, 2016), LightGBM (LGBM; Ke et al., 2017), random forest (RF; Breiman, 2001), support vector machine (SVM; Cortes & Vapnik, 1995), $L_2$-penalized logistic regression (LR), and multi-layer perceptron (MLP). We use the same 29

Table B4: Hyperparameter search space for TabR.

| Hyperparameter | Distribution |
|---|---|
| $d$ | Uniform(96,384) |
| Number of Encoder Blocks $N_E$ | Categorical([0,1]) |
| Number of Predictor Blocks $N_P$ | Categorical([1,2]) |
| Context Dropout | Uniform(0,0.6) |
| Dropout | Uniform(0,0.6) |
| Context Size | Categorical([32,64,96,128,...,512]) |
| PLR Embedding # Components | UniformInt(16,96) |
| PLR Embedding Scale $\sigma$ | LogUniform(0.01,100) |
| PLR Embedding Size | UniformInt(16,64) |
| PLR Embedding "Lite" Mode | Categorical([True,False]) |
| Learning Rate | LogUniform(1e-5,1e-2) |
| Weight Decay | LogUniform(1e-6,1e-2) |

Table B5: Hyperparameter search space for ModernNCA.

| Hyperparameter | Distribution |
|---|---|
| Linear Projection Dimension | UniformInt(64,1024) |
| MLP Block Dimension | UniformInt(64,1024) |
| Dropout | Uniform(0,0.5) |
| Number of MLP Blocks | Categorical([0,1,2]) |
| PLR Embedding # Components | UniformInt(16,96) |
| PLR Embedding Scale $\sigma$ | LogUniform(0.01,100) |
| PLR Embedding Size | UniformInt(16,64) |
| PLR Embedding "Lite" Mode | Categorical([True,False]) |
| Temperature | Categorical([0.8,0.9,1]) |
| Retrieval Sample Rate | Categorical([0,0.2,0.4,0.6,0.8,1]) |
| Learning Rate | LogUniform(1e-5,1e-2) |
| Weight Decay | LogUniform(1e-6,1e-2) |

Table B6: Hyperparameter search space for MLP-PLR.

| Hyperparameter | Distribution |
|---|---|
| Hidden Layer Dimension | UniformInt(64,1024) |
| Number of Layers | UniformInt(1,8) |
| Dropout | Uniform(0,0.5) |
| PLR Embedding # Components | UniformInt(16,96) |
| PLR Embedding Scale $\sigma$ | LogUniform(0.01,100) |
| PLR Embedding Size | UniformInt(16,64) |
| PLR Embedding "Lite" Mode | Categorical([True,False]) |
| Learning Rate | LogUniform(1e-5,1e-2) |
| Weight Decay | LogUniform(1e-6,1e-2) |

OpenML classification datasets subsampled from AMLB (Gijsbers et al., 2024), which are detailed in Extended Data Table 3 of Hollmann et al. (2025). For hyperparameter tuning, we follow the search space provided in Extended Data Table 5 for CB, XGB, and LGBM, but use alternative search spaces for the other baselines as our own search spaces (Section B) yield better results in the replicated setup (Table C1). We mainly compare the raw test AUROC values, averaged over all datasets, reported in the paper and from our own evaluations, as the min-max normalized scores in Hollmann et al. (2025) are computed with respect to TabPFN-v2 & baselines both with and without hyperparameter tuning (under a 4h budget for hyperparameter tuning), while we only we compare TabPFN-v2 (without

Table B7: Hyperparameter search space for FT-T.

| Hyperparameter | Distribution |
|---|---|
| Token Embedding Size | Categorical([64,72,80,...,512]) |
| FFN Dimension Factor | Uniform(0.67,2.67) |
| Attention Dropout | Uniform(0,0.5) |
| Residual Dropout | Uniform(0,0.2) |
| FFN Dropout | Uniform(0,0.5) |
| Number of Layers | Uniform(1,6) |
| Dropout | Uniform(0,0.5) |
| Learning Rate | LogUniform(1e-5,1e-3) |
| Weight Decay | LogUniform(1e-6,1e-3) |

Table B8: Hyperparameter search space for MLP.

| Hyperparameter | Distribution |
|---|---|
| Hidden Layer Dimension | UniformInt(64,1024) |
| Number of Layers | UniformInt(1,8) |
| Dropout | Uniform(0,0.5) |
| Learning Rate | LogUniform(1e-5,1e-2) |
| Weight Decay | LogUniform(1e-6,1e-2) |

Table B9: Hyperparameter search space for ResNet.

| Hyperparameter | Distribution |
|---|---|
| Hidden Layer Dimension | UniformInt(64,1024) |
| FFN Dimension Factor | Uniform(1,4) |
| Number of Layers | UniformInt(1,8) |
| Activation | Categorical([ReLU,GeLU,ReGLU,GeGLU]) |
| Dropout | Uniform(0,0.5) |
| Residual Dropout | Uniform(0,0.5) |
| Learning Rate | LogUniform(1e-5,1e-2) |
| Weight Decay | LogUniform(1e-6,1e-2) |

Table B10: Hyperparameter search space for MLP.

| Hyperparameter | Distribution |
|---|---|
| Hidden Layer Dimension | UniformInt(64,1024) |
| Number of Layers | UniformInt(1,8) |
| Dropout | Uniform(0,0.5) |
| Learning Rate | LogUniform(1e-5,1e-2) |
| Weight Decay | LogUniform(1e-6,1e-2) |

Table B11: Hyperparameter search space for CB.

| Hyperparameter | Distribution |
|---|---|
| Number of Estimators | UniformInt(5,125) |
| Max Depth | UniformInt(2,12) |
| Learning Rate | LogUniform(1e-5,1) |
| $L_2$ Regularization | LogUniform(1,10) |

post-hoc ensembling) against the hyperparameter-tuned baselines and do not impose an arbitrary 4-hour time limit on the hyperparameter optimization process. While there are some unresolved

Table B12: Hyperparameter search space for XGB.

| Hyperparameter | Distribution |
|---|---|
| Number of Estimators | UniformInt(5,125) |
| Max Depth | UniformInt(1,10) |
| $\alpha$ | LogUniform(1e-16,1e2) |
| $\lambda$ | LogUniform(1e-16,1e2) |
| Learning Rate | LogUniform(1e-7,1) |

Table B13: Hyperparameter search space for LGBM.

| Hyperparameter | Distribution |
|---|---|
| Number of Estimators | UniformInt(5,125) |
| Number of Leaves | UniformInt(2,4096) |
| $\alpha$ | LogUniform(1e-8,1) |
| $\lambda$ | LogUniform(1e-8,1) |
| Learning Rate | LogUniform(1e-3,1) |

Table B14: Hyperparameter search space for RF.

| Hyperparameter | Distribution |
|---|---|
| Number of Estimators | UniformInt(5,100) |
| Max Depth | UniformInt(2,12) |

Table B15: Hyperparameter search space for SVM.

| Hyperparameter | Distribution |
|---|---|
| Inverse Regularization $C$ | LogUniform(1e-5,1e5) |

Table B16: Hyperparameter search space for kNN.

| Hyperparameter | Distribution |
|---|---|
| $k$ | Categorical([3,5,7,9,…,42]) |
| NN Search | Categorical([$k$-d tree, ball tree]) |
| NN Weighting | Categorical([Uniform, Inverse-Distance]) |

Table B17: Hyperparameter search space for LR (Logistic Regression).

| Hyperparameter | Distribution |
|---|---|
| Inverse Regularization $C$ | LogUniform(1e-10,1e10) |

Table B18: Hyperparameter search space for LR (Ridge Regression).

| Hyperparameter | Distribution |
|---|---|
| Regularization $\alpha$ | LogUniform(1e-10,1) |

differences between the reported numbers and our own, partially due to the lack of access to the exact code that was used by Hollmann et al. (2025), they are similar across the board for most models, with some baseline numbers even being better than the reported numbers (e.g., MLP and LR).

Table C1: Comparison of average test AUROCs between the reported numbers and our numbers, in our replication of the evaluation setup considered in Hollmann et al. (2025). We report the numbers for Hollmann et al. (2025) based on Extended Data Table 5, where for TabPFN-v2, we report the average test AUROC corresponding to "TabPFN (default)", and for the baselines, we report the average test AUROC corresponding to the "4h tuned" models.

|  | TabPFN-v2 | CB | XGB | LGBM | RF | MLP | SVM | LR |
|---|---|---|---|---|---|---|---|---|
| Hollmann et al. (2025) | 0.929 | 0.920 | 0.920 | 0.915 | 0.913 | 0.883 | 0.887 | 0.874 |
| **Ours** | 0.928 | 0.912 | 0.920 | 0.911 | 0.905 | 0.897 | 0.877 | 0.881 |

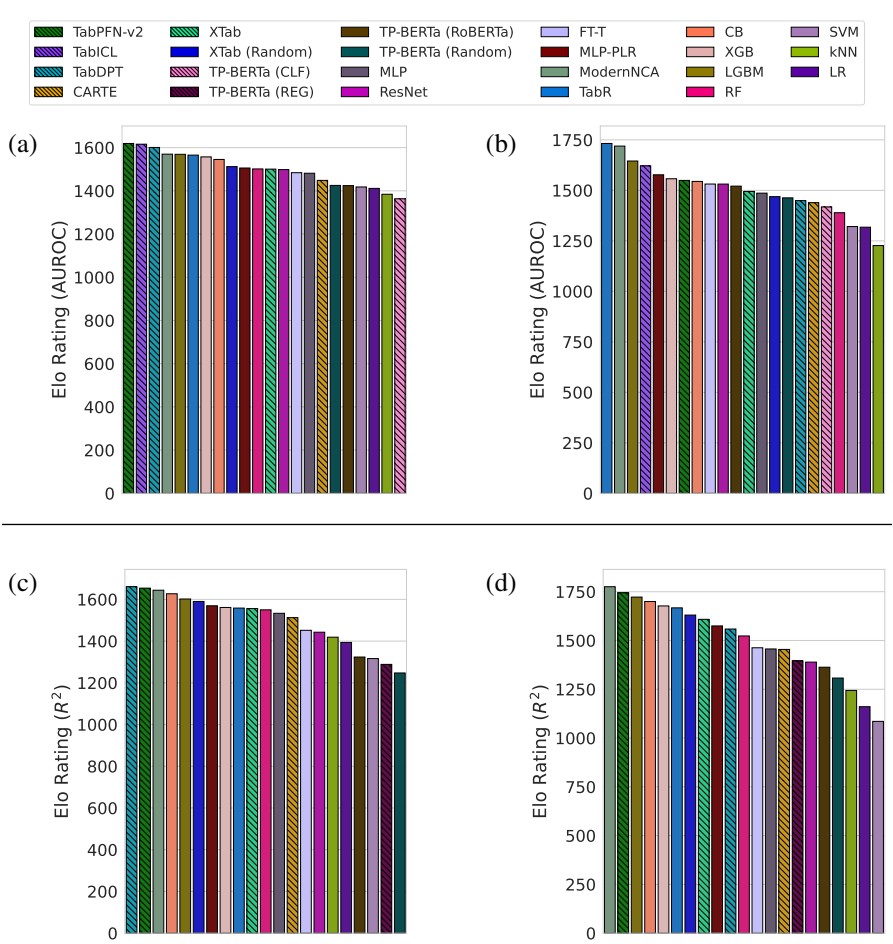

Figure D1: In the full-data setting, the performance gap between TFMs and non-pretrained baselines on classification and regression tasks remain small, across both small-data and large-data regimes. Here, we show the Elo ratings computed based on the 95% bootstrapping confidence intervals in relative AUROC / $R^2$ (Section 3). (a) Classification results on small datasets in the "TabPFN" regime. (b) Classification results on large datasets. (c) Regression results on small datasets in the "TabPFN" regime. (d) Regression results on large datasets.

## D  ADDITIONAL EXPERIMENTAL RESULTS

Here, we provide additional results for the experiments discussed in Section 4.

**Evaluations in the full-data setting (Section 4).** In Figures D1(a) and D1(b), we show the Elo rating plots on the classification datasets in the small "TabPFN" regime and the alternative large-data

regime, respectively. In Figures D1(c) and D1(d), we show the same plots for the regression datasets. Across all datasets, we find that ModernNCA is the best-performing non-pretrained baseline (i.e., with the highest Elo rating), and therefore the pairwise win/tie/loss rates (%) for each TFM shown in Figure 2 are calculated with respect to ModernNCA.

In Figures D2–D5, we also show the (i) win, tie, and loss rates (%) of each TFM vs. the best-performing baseline; (ii) the critical difference diagrams along with Nemenyi-Friedman post-hoc test results; and (iii) the Elo ratings calculated based on the 95% bootstrapping CIs, under several alternative stratifications of the datasets. In particular, we consider splitting the datasets by each criterion below:

1. Column names are grounded on real-world concepts and meaningful (e.g., "blood pressure");

2. More than 50% of features are categorical (i.e., dataset is "highly categorical");

3. Dataset contains features with missing values[3];

4. Ratio of number of samples in the minority class to that in the majority class is less than 50% (i.e., dataset has high class imbalance)[4].

Among many other possible stratifications, we consider these four settings as a representative set of alternative scenarios where TFMs are often claimed to better handle or expected to perform better (Kim et al., 2024; Gardner et al., 2024; Hollmann et al., 2025) (e.g., models that leverage LLM embeddings of column names are expected to perform better when the dataset is rich with meaningful column names). For each stratification, we compute the win, tie, and loss rates (%) for each TFM with respect to the best-performing baseline with the highest Elo rating. Overall, the performance gains achieved by each TFM over non-pretrained baselines remain small across most of these settings, even for tables with meaningful column names, where language-based TFMs are supposed to provide a decisive advantage.

**Evaluations in the few-shot setting (Section 4).** In Figures D6(a) and D6(b), we show the Elo rating plots on the classification datasets in the 8-shot and 32-shot settings. In Figures D6(c) and D6(d), we show the same plots for the regression datasets. As all of the few-shot evaluations are repeated with 5 different random seeds, which control the subsampling of the $k$-shot examples used for training and validation, we compute the Elo ratings for each seed and then average the Elo ratings. As discussed in Section 4, we observe that the performance gap between TFMs and non-pretrained baselines on classification and regression tasks remain small for most models in the few-shot setting.

# E  LIMITATIONS

We discuss our findings with the following caveats. First, our evaluations are primarily focused on the *performance* benefits from large-scale tabular pretraining, and therefore our analysis and conclusions about the utility of such pretraining do not account for improvements in off-the-shelf training and inference runtimes, which several prior works highlight as a major practical advantage (Hollmann et al., 2023; 2025; Qu et al., 2025). Second, we deliberately excluded post-hoc ensembling for all of the TFMs to isolate the impact of large-scale tabular pretraining as much as possible and to ensure a fair comparison among all TFMs, although it is often considered for ICL models like TabPFN-v2. As such, the resulting numbers may not be reflective of the best performance that can be achieved by these models on the datasets we considered. Third, while we have sought to include in our evaluation a diverse set of open-source TFMs—excluding those for which the checkpoints or relevant code are unavailable—our findings may not be characteristic of all existing TFMs that have yet to be included in our experiments (e.g., hypernetwork models (Müller et al., 2025)). It is certainly possible that newly released TFMs, pretrained at even larger scales and with improved inductive biases, do consistently show statistically significant performance gains across various settings, and we leave an extended analysis of any additional up-to-date models as future work. Last but not least, while our benchmark was carefully designed to include datasets with a wide range of characteristics while ensuring that they are challenging enough to reflect the difficulties of real-world prediction tasks,

---

[3]We group all datasets with missingness together as there are not as many to further divide into subgroups with different levels of missingness.

[4]We choose 50% as the threshold such that we divide the datasets by roughly half.

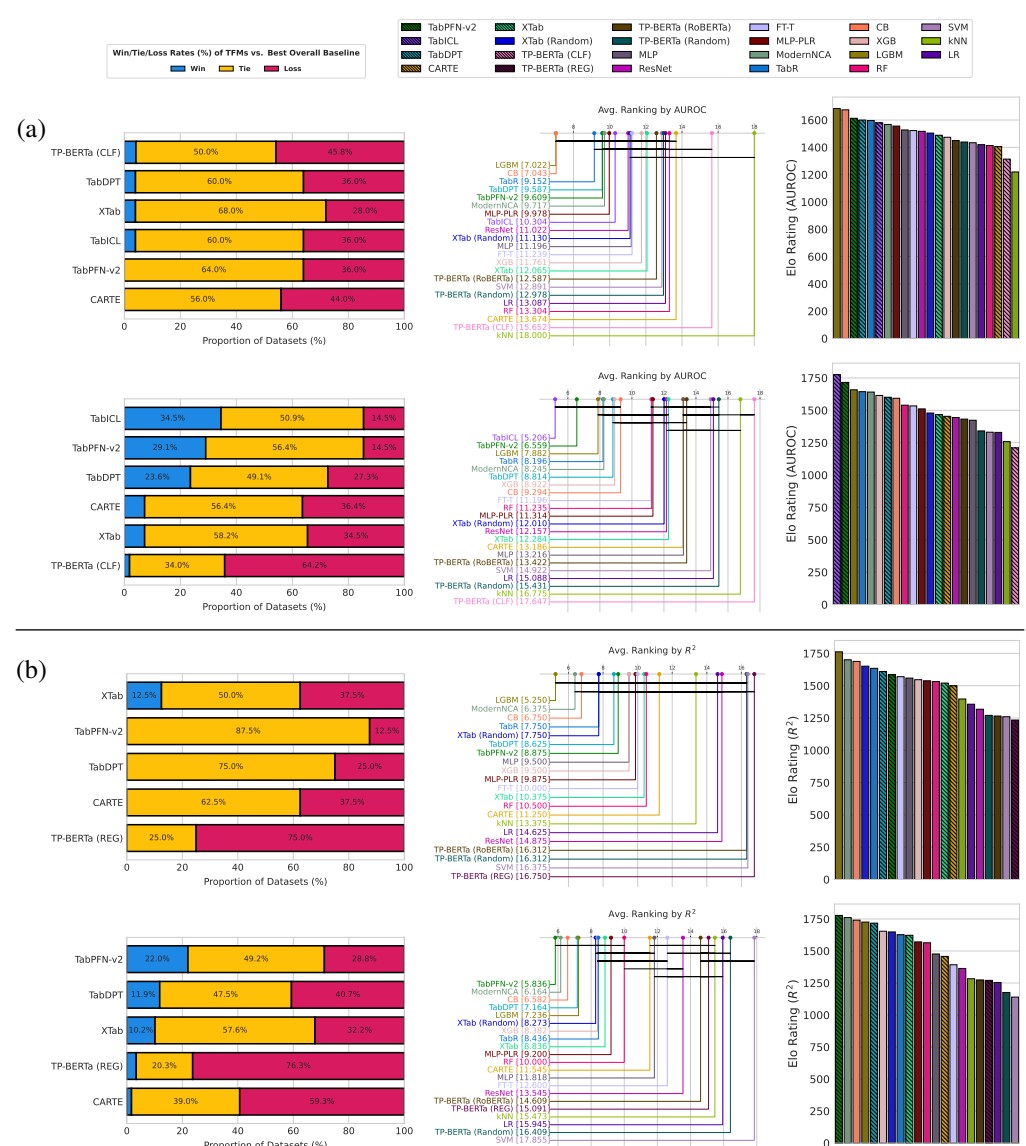

Figure D2: In the full-data setting (i.e., using all training examples), TFMs show modest improvements over non-pretrained baselines on both "highly categorical" datasets (i.e., at least 50% of features are categorical) and "highly numerical" datasets. Panel (a) shows classification tasks, and panel (b) shows regression tasks. In each panel, the top row shows the results on the "highly categorical" datasets, while the bottom row shows the results on the "highly numerical" datasets. (Left) Pairwise win, tie, and loss rates (%) of each TFM vs. the baseline with the highest Elo rating (Appendix D). On each dataset, a TFM "wins" over the best-performing baseline if the 95% bootstrapping confidence interval (CI) in their relative AUROC / $R^2$ lies above 0 (Section 3). (Middle) Critical difference diagram based on the Nemenyi-Friedman test results. If two models are connected horizontally, their differences in ranking are *not* statistically significant. (Right) Elo ratings of all models computed based on the 95% bootstrapping CIs.

our analysis is still limited to OpenML datasets, which dominate most existing tabular prediction benchmark studies and still may not be reflective of realistic settings (Rubachev et al., 2024). As such, we also leave an extension of our analysis to *non-OpenML* datasets with clearly verifiable real-world sources (e.g., electronic health record datasets such as MIMIC (Johnson et al., 2016; 2023) or eICU (Pollard et al., 2018), industry-grade datasets on Kaggle) as future work.

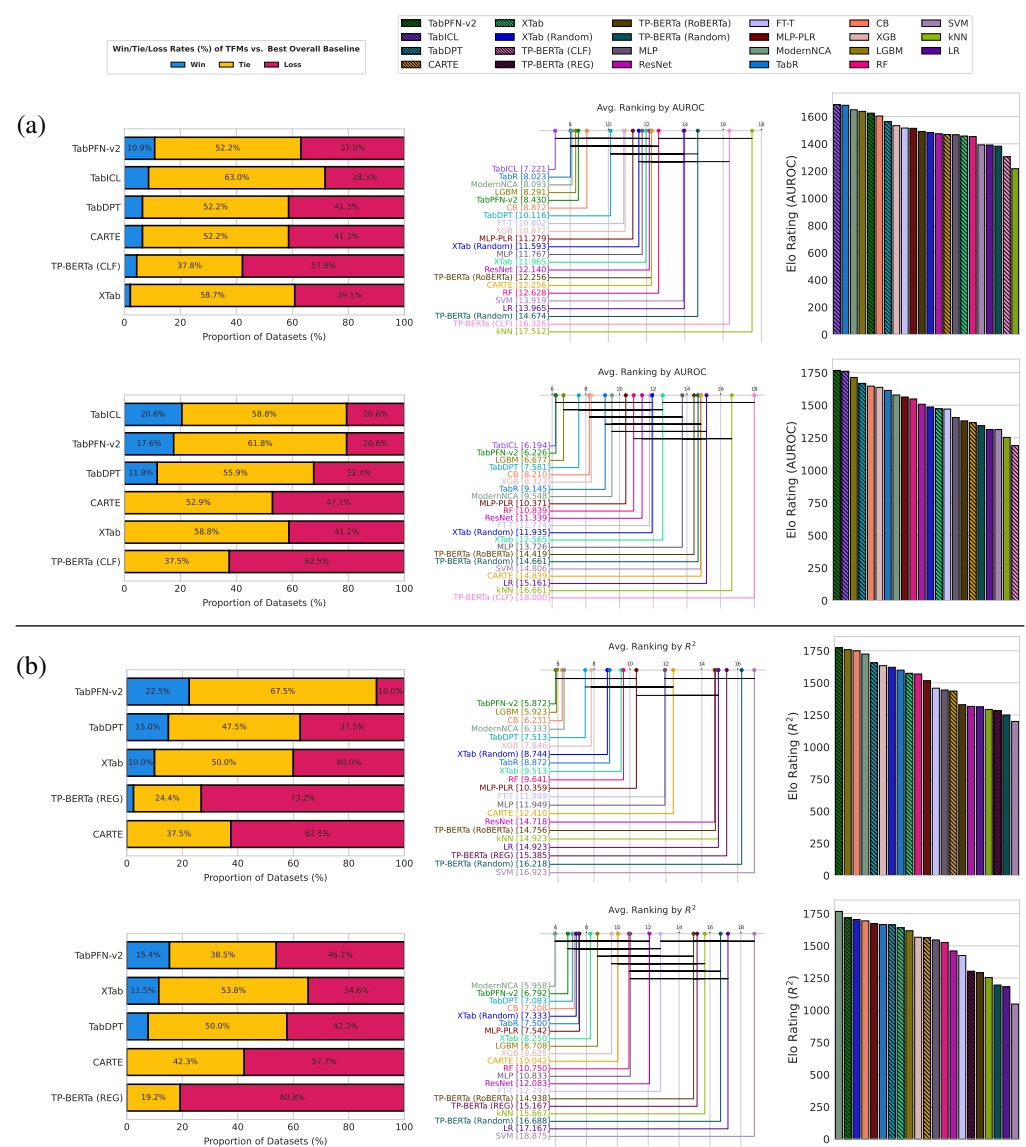

Figure D3: In the full-data setting (i.e., using all training examples), TFMs show modest improvements over non-pretrained baselines on both datasets with and without meaningful column names. Panel (a) shows classification tasks, and panel (b) shows regression tasks. In each panel, the top row shows the results on datasets with meaningful column names, while the bottom row shows the results on the datasets without. (Left) Pairwise win, tie, and loss rates (%) of each TFM vs. the baseline with the highest Elo rating (Appendix D). On each dataset, a TFM "wins" over the best-performing baseline if the 95% bootstrapping confidence interval (CI) in their relative AUROC / $R^2$ lies above 0 (Section 3). (Middle) Critical difference diagram based on the Nemenyi-Friedman test results. If two models are connected horizontally, their differences in ranking are *not* statistically significant. (Right) Elo ratings of all models computed based on the 95% bootstrapping CIs.

## F    SOURCE CODE

An anonymized version of our source code can be found here: https://anonymous.4open.science/r/tfm-evaluation-harness-7D90/.

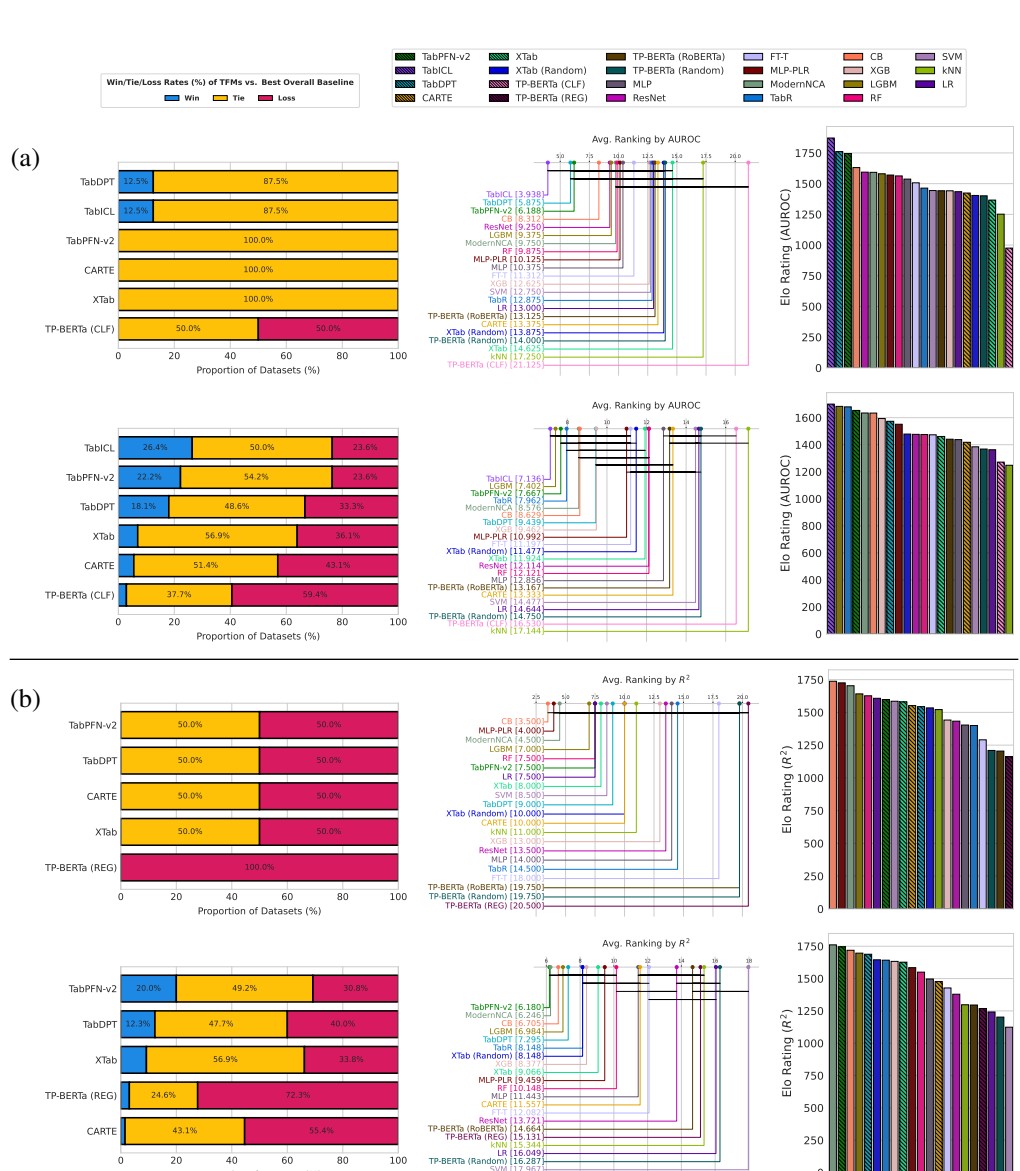

Figure D4: In the full-data setting (i.e., using all training examples), TFMs show modest improvements over non-pretrained baselines on both datasets with and without features with missing values. Panel (a) shows classification tasks, and panel (b) shows regression tasks. In each panel, the top row shows the results on datasets with missing values, while the bottom row shows the results on the datasets without. (Left) Pairwise win, tie, and loss rates (%) of each TFM vs. the baseline with the highest Elo rating (Appendix D). On each dataset, a TFM "wins" over the best-performing baseline if the 95% bootstrapping confidence interval (CI) in their relative AUROC / $R^2$ lies above 0 (Section 3). (Middle) Critical difference diagram based on the Nemenyi-Friedman test results. If two models are connected horizontally, their differences in ranking are *not* statistically significant. (Right) Elo ratings of all models computed based on the 95% bootstrapping CIs. Meanwhile, we note that there are only 2 datasets with missing values for regression (Figure D4(b), top), which results in a failure to reject the null for all pairwise comparisons performed with Nemenyi-Friedman.

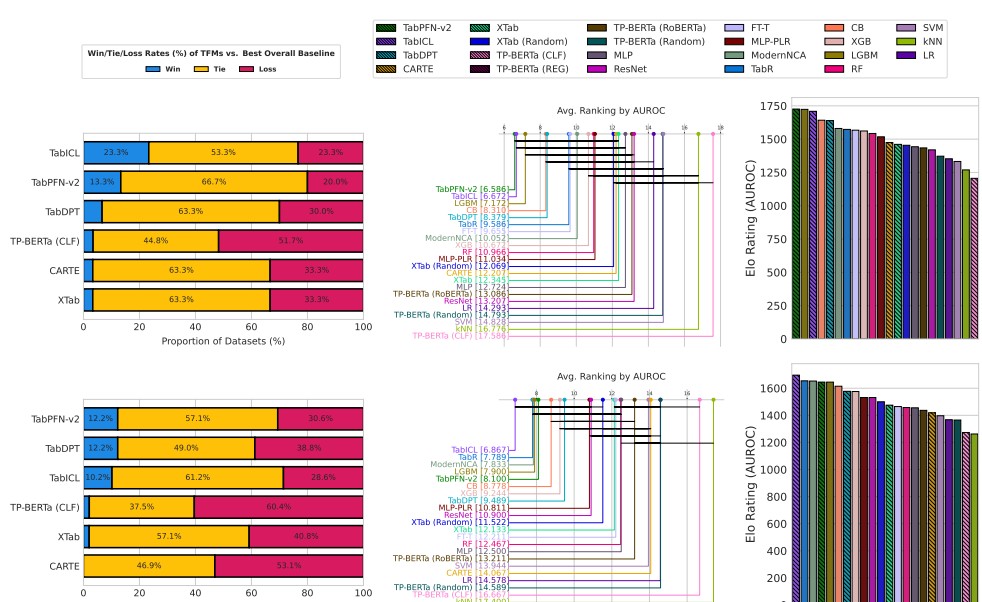

Figure D5: In the full-data setting (i.e., using all training examples), TFMs show modest improvements over non-pretrained baselines on classification datasets with and without class imbalance. The top row shows the results on datasets with class imbalance, while the bottom row shows the results on the datasets without. (Left) Pairwise win, tie, and loss rates (%) of each TFM vs. the baseline with the highest Elo rating (Appendix D). On each dataset, a TFM "wins" over the best-performing baseline if the 95% bootstrapping confidence interval (CI) in their relative AUROC / $R^2$ lies above 0 (Section 3). (Middle) Critical difference diagram based on the Nemenyi-Friedman test results. If two models are connected horizontally, their differences in ranking are *not* statistically significant. (Right) Elo ratings of all models computed based on the 95% bootstrapping CIs.

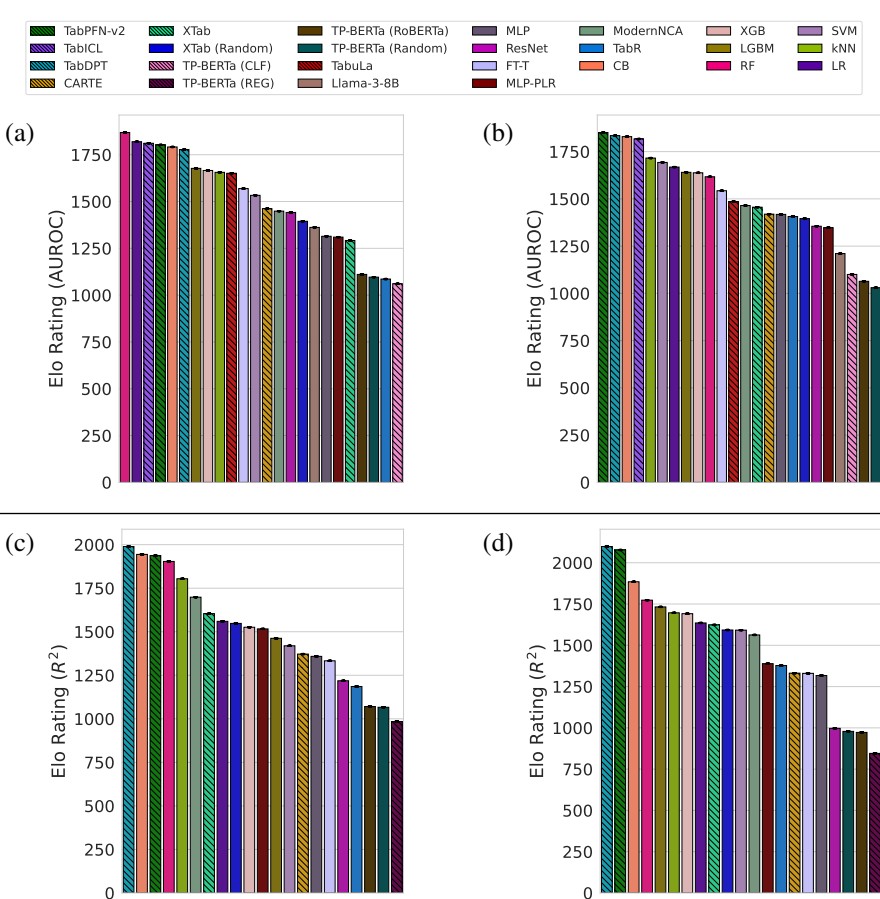

Figure D6: The performance gap between TFMs and non-pretrained baselines on classification and regression tasks remain small for most models in the few-shot setting. Here, we show the Elo ratings computed based on the 95% bootstrapping confidence intervals in relative AUROC / $R^2$ (Section 3). (a) Classification results in the 8-shot setting. (b) Classification results in the 32-shot setting. (c) Regression results in the 8-shot setting. (d) Regression results in the 32-shot setting. As discussed in Section 4, we do find that TabPFN-v2 and TabDPT tend to show better sample efficiency on *regression* tasks, while all other TFMs do not. Error bars indicate the standard error across datasets.

