# OpenReview forum: "Large-Scale Pretraining Offers Modest Benefits for Tabular Transfer Learning"
_ICLR.cc/2026/Conference — Submitted to ICLR 2026_

### Official Review · Reviewer_97fx · 2025-10-27

**Soundness:** 4
**Presentation:** 4
**Contribution:** 3
**Rating:** 6
**Confidence:** 5

**Summary:**

This paper examines the tabular foundation models (TFMs) and their claims to be better than tree-based baselines or the non-pretrained baselines. The authors used comprehensive experiments to show that some of the claims TFM papers made were slightly misleading. The authors used the original scale and tested for statistical significance and found a different conclusion. The paper concluded that TFMs only offer modest performance gain even from large scale pre-training.

**Strengths:**

### Originality and Significance
1. The main claim of the paper is original and has not been discussed rigorously in the past
2. The paper offers comparison of most popular and recent TFMs which other studies have not done

### Quality and Clarity
1. The paper is well written and very clear throughout
2. The experiments are well designed. The use of bootstrapping for statistical significance is useful.

**Weaknesses:**

1. The paper has not examined the computational complexity of each method.
* The overhead of selecting hyper-parameters for baseline methods might be higher than the TFM methods if not considering the pre-training cost of TFMs.
2. There is no study on fine-tuning or other adaptation methods such as feature ensembling used by TabPFN

**Questions:**

1. Is the validation set passed to TFMs as context?
2. Do the TFM models fail to the baseline on the same datasets?
3. Will there be a rule of thumb for choosing which method to use based on the dataset?

---

> ### Author Response · Authors · 2025-11-25
> **Author Response**
>
> Thank you for your detailed feedback and thoughtful questions. We are glad to see that the reviewer found our paper as presenting “original” insights that “[have] not been discussed rigorously in the past” based on “comparison[s] of most popular and recent TFMs”, with “experiments that are well designed” and a write-up that is “well written and very clear throughout”. Below, we provide our response to the individual questions and concerns raised by the reviewer.
>
> > “The overhead of selecting hyper-parameters for baseline methods might be higher than the TFM methods if not considering the pre-training cost of TFMs.”
>
> > “The paper has not examined the computational complexity of each method.”
>
> Thank you for the feedback. As our paper is focused on the *performance* benefits from tabular pretraining, we mention the lack of discussions on computational efficiency as a limitation of our study in Appendix E (Limitations):
>
> “...our evaluations are primarily focused on the *performance* benefits from large-scale tabular pretraining, and therefore our analysis and conclusions about the utility of such pretraining do not account for improvements in off-the-shelf training and inference runtimes, which several prior works highlight as a major practical advantage (Hollmann et al., 2023; 2025; Qu et al., 2025).”
>
> > “There is no study on fine-tuning or other adaptation methods such as feature ensembling used by TabPFN”
>
> We would like to clarify a possible misunderstanding. We *do not* turn off the *feature preprocessing steps* considered by the tabular ICL models (TabPFN-v2, TabICL, TabDPT). The ICL models make each test prediction by aggregating the outputs from multiple forward passes with different feature preprocessing steps, and these steps are still present.
>
> The only thing that we turn off is the post-hoc ensembling approach proposed for TabPFN-v2, which involves ensembling *multiple* TabPFN-v2 models as opposed to using just one instantiation of the model.
>
> To avoid any confusion, in our revised draft, we have updated the “Tabular foundation models (TFMs)” paragraph in Section 3 (**in blue**) to:
>
> “TabPFN-v2 can be evaluated with *post-hoc ensembling* (PHE; detailed in the “TabPFN (PHE)” section of Hollmann et al. (2025)), where each prediction is obtained by aggregating the outputs from *multiple* instantiations of TabPFN-v2 with greedy ensemble selection (Caruana et al., 2004). However, we *intentionally exclude PHE* for TabPFN-v2 to (i) ensure a consistent setup with other TFMs which are not post-hoc ensemble in the original paper and (ii) isolate the impact of tabular pretraining on downstream performance as much as possible. **The “default” preprocessing steps and inference configuration for TabPFN-v2 (see Extended Data Table 5 of Hollmann et al. (2025)) are still internally applied within the model’s forward pass.**”
>
> > “Is the validation set passed to TFMs as context?”
>
> No, they are not. For each dataset, we only provide the training dataset as context to the TFMs. This is to ensure that all models are “fit” only on the training dataset, for an apples-to-apples comparison.
>
> > “Do the TFM models fail to the baseline on the same datasets?”
>
> No, different TFMs fail on different datasets. Using TabPFN-v2, TabICL, TabDPT, and CARTE as an example, we show the proportion of classification datasets where each pair of TFMs show the same trends (i.e., win, tie, loss) when compared to CatBoost:
>
> | TFM 1 | TFM 2 | Proportion of datasets where comparisons against CatBoost match (%) |
> | - | - | - |
> | TabPFN-v2 | TabICL | 79.3% |
> | TabPFN-v2 | TabDPT | 53.7% |
> | TabPFN-v2 | CARTE  | 56.1% |
> | TabICL | TabDPT | 51.8% |
> | TabICL | CARTE | 50.0% |
> | TabDPT | CARTE | 50.0% |
>
> > “Will there be a rule of thumb for choosing which method to use based on the dataset?”
>
> Given the heterogeneity of tabular datasets, it is challenging to identify a simple generalizable rule for determining which TFM is most appropriate for a particular dataset simply based on the dataset's characteristics.
>
> Meanwhile, we do observe that tabular ICL models tend to perform better on small classification datasets in the full-data setting (Finding 2, Section 4). In the other data regimes we consider for full-data evaluations, we do not observe a consistent pattern across TFMs for when they outperform the non-pretrained baselines (Table 1). In the 8-shot and 32-shot settings, we find that TabPFN-v2 and TabDPT show some improvements in sample efficiency on small regression datasets (Finding 3, Section 4).

---

> > ### Comment · Area_Chair_UGxi · 2025-11-28
> > **Reminder to Respond to Rebuttal**
> >
> > Reviewer 97fx,
> >
> > Please ensure that you read the authors' response to your review and post an acknowledgement **before end of Dec. 2**. This date is the cutoff for reviewer posts.
> >
> > Thank you for supporting quality peer review at ICLR.
> >
> > AC

---

### Official Review · Reviewer_Uc6Q · 2025-11-01

**Soundness:** 3
**Presentation:** 3
**Contribution:** 3
**Rating:** 6
**Confidence:** 4

**Summary:**

The authors evaluate a range of tabular foundation models with a focus on critically evaluating performance differences between methods, challenging previous papers that showed TFMs significantly outperforming baselines. They argue that their results suggest limited benefit to large-scale tabular pretraining.

**Strengths:**

- The paper points to a common shortcoming of the tabular modelling literature in that metrics are preferentially chosen that emphasize the difference between models which often don't have large differences in absolute performance.
- Having an independent and critical evaluation of tabular models is a useful contribution.
- A wide range of interesting experiments are provided with applicability to different tabular settings.

**Weaknesses:**

- Most tabular foundation models under evaluation don't perform large-scale pretraining in a way that is comparable to vision and language models, making the framing of the paper somewhat overstated. TabDPT, XTab, and TP-BERTa don't perform large-scale pretraining at all, only using 123, 202, and 52 tables for pretraining respectively. TabPFNv2 and TabICL were trained at scale but only on synthetic data. CARTE pretraining is on graph relationships rather than tabular data.
- The core point of the paper is rather subtle and it could be presented with more rigour. The authors refer to "the statistical significance of the extent of improvement" and "whether the absolute gains in performance themselves are statistically significant", but this differs from the standard usage of statistical significance - either differences are statistically significant at some level or they aren't. The paper prefers certain approaches for evaluating statistical significance but it's not clear that this invalidates the statistical significance evaluations in existing tabular modelling papers.
- The evaluations deliberately skip preprocessing hyperparameter tuning for in-context learning models where available but perform extensive hyperparameter tuning for other models, which could account for some of the reduced gap in performance. I'm not convinced this is fair, and it wouldn't be applicable to a practitioner seeking the most performant model.

**Minor**

- Typo line 238: "classicial"

**Questions:**

- Could you clarify how the "best baseline" was selected in results that include it? If this was selected based on the test data that's being shown, then it effectively means model selection on test data, which isn't a statistically fair comparison.
- What is the interpretation of models showing statistically significant differences when min-max scaling is applied but not when it isn't? These both seem like valid comparisons for testing significance, even if differences are more or less emphasized. Similarly, how should we interpret a significant difference being shown by a Wilcoxon signed-rank test but not by this paper's bootstrapping procedure?
- In your "non-tabular" exclusion, how did you treat datasets that originally come from non-tabular data such as images, but used extraction methods to produce numeric summary features?

---

> ### Author Response · Authors · 2025-11-25
> **Author Response [1/3]**
>
> Thank you for your detailed feedback and thoughtful suggestions. We are glad to see that the reviewer found our paper as presenting “a wide range of interesting experiments” “with applicability to different tabular settings” and a “useful contribution” that provides “an independent and critical evaluation of tabular models”, demonstrating the “common shortcoming of the tabular modeling literature”. Below, we provide our response to the individual questions and concerns raised by the reviewer.
>
> > “Most tabular foundation models under evaluation don’t perform large-scale pretraining in a way that is comparable to vision and language models, making the framing of the paper somewhat overstated.”
>
> We do not intend to claim that “tabular pretraining can never work”, i.e., that there is an inherent ceiling to the performance benefits from tabular pretraining. It is certainly possible that, at larger scales and with a different style of pretraining, TFMs consistently show significant performance improvements over non-pretrained baselines.
>
> Rather, we critically evaluate the widespread belief that “tabular pretraining currently works well”. All prior works on TFMs claim substantial improvements over non-pretrained baselines, while our findings suggest that these claims might be *overly optimistic*.
>
> Meanwhile, to address this concern, we have added the following (**in orange**) to our Limitations section (Appendix E):
>
> “It is certainly possible that newly released TFMs, pretrained at even larger scales and with improved inductive biases, do consistently show statistically significant performance gains across various settings, and we leave an extended analysis of any additional up-to-date models as future work.”
>
> > “TabDPT, XTab, and TP-BERTa don’t perform large-scale pretraining at all… TabPFNv2 and TabICL were trained at scale but only on synthetic data. CARTE pretraining is on graph relationships rather than on tabular data.”
>
> We agree that the scale and “form” of pretraining considered in the tabular domain have clear differences with those considered in vision and language. Nonetheless, all of the models that we evaluate share the same motivation: to improve performance on a downstream tabular dataset via pretraining on many other tables. We sought to evaluate as many such models as available (at the time of writing), focusing on those that claim strong performance and well-recognized by the community.
>
> While TabPFN-v2 and TabICL are indeed only pretrained on synthetic data, both works are advertised by the authors as a general-purpose “foundation model”, and we believe that it is fair to categorize them as such. TabDPT is also advertised by the authors as a general-purpose “foundation model”, despite the relatively limited scale.
>
> For CARTE, we interpret its pretraining procedure on large relational databases (YAGO3) as a method for transferring knowledge from relational “tables” (as a knowledge graph can also be interpreted as a table of triplets) to improve the predictive performance on other downstream tabular datasets. XTab and TP-BERTa were not trained at scale but nonetheless claim significant performance gains from training on other tables.

---

> ### Author Response · Authors · 2025-11-25
> **Author Response [2/3]**
>
> > “The core point of the paper is rather subtle and it could be presented with more rigour… The paper prefers certain approaches for evaluating statistical significance but it’s not clear that this invalidates the statistical significance evaluations in existing tabular modelling papers.”
>
> > “What is the interpretation of models showing statistically significant differences when min-max scaling is applied but not when it isn’t? These both seem like valid comparisons for testing significance, even if differences are more or less emphasized. Similarly, how should we interpret a significant difference being shown by a Wilcoxon signed-rank test but not by this paper’s bootstrapping procedure?”
>
> Our bootstrapping procedure evaluates improvements at a *per-dataset* level, and the resulting win/tie/loss rates quantify “how often does a TFM outperform the best baseline by a statistically significant margin?” In contrast, the Wilcoxon signed-rank test asks “does a TFM tend to rank higher than the best baseline *across all datasets*?”.
>
> The latter statistical question is indeed also valid and relevant, but the Wilcoxon test does not test whether the pairwise performance difference (e.g., $\Delta$AUROC) observed on each dataset is large enough to be statistically meaningful. In other words, it does not account for the “effect size” of improvement.
>
> As such, it is entirely possible for ranking-based tests like Wilcoxon to report significance even when the underlying per-dataset performance gaps are within the noise level. If the Wilcoxon test is significant but our bootstrapping procedure is not (e.g., high tie rates, with similar win and loss rates), then this indicates that a TFM consistently ranks higher, but the average “effect size” of performance improvement across datasets is very small.
>
> Meanwhile, min-max scaling can inflate very small pairwise performance differences, e.g., on datasets where all models achieve similar performance (Figure 1). The Wilcoxon signed-rank test statistic is computed as $T = T^+ - T^-$, where $T^+$ and $T^-$ sum the ranks of the absolute values of positive (i.e., TFM > baseline) and negative (TFM < baseline) per-dataset differences, respectively, and the test measures how much $T$ deviates from 0. Min-max scaling can alter the rankings of absolute pairwise differences, which may strengthen the significance results from the Wilcoxon test even though the raw performance differences are actually small.
>
> > “The evaluations deliberately skip preprocessing hyperparameter tuning for in-context learning models”
>
> We would like to clarify a possible misunderstanding. We *do not* turn off the *feature preprocessing steps* considered by the tabular ICL models (TabPFN-v2, TabICL, TabDPT). The ICL models make each test prediction by aggregating the outputs from multiple forward passes with different feature preprocessing steps, and these steps are still present.
>
> More precisely:
> - TabPFN-v2: We use the “default” feature preprocessing steps recommended by the authors in Extended Data Table 5 of [1], which are internal to the model’s forward pass.
> - TabDPT: We use the same feature preprocessing steps provided by the authors in Appendix C.1 of [2].
> - TabICL: We use the same feature preprocessing steps provided by the authors in Section 5.1 of [3].
>
> These preprocessing steps are *internally* applied (i.e., within the forward pass of each ICL model), on top of the data that has already been preprocessed as detailed in the “Datasets” paragraph of Section 3. **As these settings reflect what was considered by the authors in their original evaluations, we believe that this is a fair representation of these models.**
>
> The only part that we turn off is the post-hoc ensembling approach proposed for TabPFN-v2, which involves ensembling *multiple* TabPFN-v2 models as opposed to using just one instantiation of the model.
>
> To avoid any confusion, in our revised draft, we have updated the “Tabular foundation models (TFMs)” paragraph in Section 3 (**in blue**) to:
>
> “TabPFN-v2 can be evaluated with *post-hoc ensembling* (PHE; detailed in the “TabPFN (PHE)” section of Hollmann et al. (2025)), where each prediction is obtained by aggregating the outputs from *multiple* instantiations of TabPFN-v2 with greedy ensemble selection (Caruana et al., 2004). However, we *intentionally exclude PHE* for TabPFN-v2 to (i) ensure a consistent setup with other TFMs which are not post-hoc ensemble in the original paper and (ii) isolate the impact of tabular pretraining on downstream performance as much as possible. **The “default” preprocessing steps and inference configuration for TabPFN-v2 (see Extended Data Table 5 of Hollmann et al. (2025)) are still internally applied within the model’s forward pass.**”

---

> > ### Author Response · Authors · 2025-11-25
> > **Author Response [3/3]**
> >
> > > “Could you clarify how the “best baseline” was selected in results that include it? If this was selected based on the test data that’s being shown, then it effectively means model selection on test data, which isn’t a statistically fair comparison.”
> >
> > We would like to clarify a possible misunderstanding. The “best baseline” was *not* adversarially selected on a *per-dataset* basis. Instead, this baseline corresponds to the non-pretrained model that had the highest Elo rating (see e.g., captions for Figures 2 and 3), which is computed by *aggregating the results across all datasets and model pairs* (see Section 2).
> >
> > For example, in Figure 2(a), all of the win-tie-loss rate plots are between each TFM and *ModernNCA*, which had the highest Elo rating among the baselines, for evaluations on small classification datasets in the “TabPFN” regime (see second paragraph of Finding 2 (Section 3) and Appendix D).
> >
> > > “In your “non-tabular” exclusion, how did you treat datasets that originally come from non-tabular data such as images, but used extraction methods to produce numeric summary features?”
> >
> > We only excluded non-tabular datasets that were provided in their raw format (e.g., Fashion-MNIST, where the raw pixel values were flattened to form the input feature vectors). For the datasets that provided the summary statistics as opposed to the non-tabular features in their raw format, we included them into the benchmark. We have updated the corresponding text in Section 3 to clarify this description (as a footnote, **in brown**):
> >
> > “We still include datasets that are originally from “non-tabular'' data but contain tabular summary statistics.”

---

> > > ### Author Response · Authors · 2025-11-25
> > > **Author Response [References Discussed in Above Responses]**
> > >
> > > **References:**
> > >
> > > [1] Accurate Predictions on Small Data with a Tabular Foundation Model (Hollmann et al., 2025)
> > >
> > > [2] TabDPT: Scaling Tabular Foundation Models on Real Data (Ma et al., 2025)
> > >
> > > [3] TabICL: A Tabular Foundation Model for In-Context Learning on Large Data (Qu et al., 2025)

---

> > > > ### Comment · Reviewer_Uc6Q · 2025-11-28
> > > >
> > > > Thank you for the response. I'd like to clarify W3:
> > > >
> > > > My concern was not that preprocessing had been turned off, but that hyperparameter optimization for preprocessing and for other TFM hyperparameters was not conducted, while extensive hyperparameter tuning was conducted on the baseline models, which does not strike me as fair. The TabArena benchmark provides a good example of TFMs being evaluated with hyperparameter tuning.
> > > >
> > > > Your previous revision explicitly noted that such tuning was not done: "While TabPFN-v2 and TabICL are often evaluated after optimizing their internal feature preprocessing steps [...] we intentionally exclude these steps." I think this acknowledgment should have been left in for transparency, unless it is incorrect.

---

> > > > > ### Author Response · Authors · 2025-12-03
> > > > > **Author Response**
> > > > >
> > > > > > “Thank you for the response. I'd like to clarify W3:
> > > > > My concern was not that preprocessing had been turned off, but that hyperparameter optimization for preprocessing and for other TFM hyperparameters was not conducted, while extensive hyperparameter tuning was conducted on the baseline models, which does not strike me as fair. The TabArena benchmark provides a good example of TFMs being evaluated with hyperparameter tuning.”
> > > > >
> > > > > Thank you for the thoughtful response and clarification. However, we disagree with the reviewer’s perspective that our evaluation approach systematically disadvantages tabular ICL models.
> > > > >
> > > > > **First**, we note that TabArena [1] also evaluates all tabular ICL models except TabPFN-v2 without additional model tuning. In Section 2.1 of [1], the authors of TabArena state:
> > > > >
> > > > > *“TabICL and TabDPT do not specify hyperparameter optimization (HPO) in the original paper and implementation; thus, we restrict ourselves to only evaluating their default performance.”*
> > > > >
> > > > > *Just like TabArena*, we use the default inference configuration recommended and used by the *original authors* of TabICL (see Section 5.1 of [2]) and TabDPT (see Appendix C.1 of [3]). In Section 2.1 of [1], the authors of TabArena also state that “every included model… was implemented in dialogue with authors”. As such, we believe that this is a fair setup.
> > > > >
> > > > > **Second**, for TabPFN-v2, we indeed use the “default” inference configuration—detailed in Extended Data Table 5 of [4]—without **(i)** additionally tuning the feature preprocessing steps or **(ii)** post-hoc ensembling (PHE), which are the two main “hyperparameters” noted by the reviewer.
> > > > >
> > > > > However, regarding **(i)**, we note that the feature preprocessing steps included in the “default” inference configuration are *already "tuned"* (just not on a per-dataset basis). In the “Default configuration of TabPFN” section of [4], the authors of TabPFN-v2 state:
> > > > >
> > > > > *“Our default configuration (TabPFN (default)) for both classification and regression is optimized for accurate predictions with minimal fitting time… The settings for our data processing were obtained through a hyperparameter search optimized on our development datasets. The exact settings chosen are listed in Extended Data Table 5.”*
> > > > >
> > > > > This “default” configuration is also the setup that was used in the main evaluations by the original authors [4] and other follow-up works like [2]. Therefore, while we acknowledge that additional tuning of feature preprocessing steps can lead to better results for TabPFN-v2, we disagree with the perspective that our setup significantly disadvantages TabPFN-v2.
> > > > >
> > > > > Regarding **(ii)**, we deliberately excluded PHE for TabPFN-v2 for an *apples-to-apples* comparison: none of the other TFMs or baselines undergo PHE, and it would be unfair to use PHE for only TabPFN-v2. *Even in TabArena*, either all models are post-hoc ensembled for comparison (“tuned + ensembled” setting) or not (“default” or “tuned” setting). While extending the analysis to the PHE setting may be an interesting addition to our study, we again disagree with the perspective that our setup significantly disadvantages TabPFN-v2.
> > > > >
> > > > > > “Your previous revision explicitly noted that such tuning was not done: "While TabPFN-v2 and TabICL are often evaluated after optimizing their internal feature preprocessing steps [...] we intentionally exclude these steps." I think this acknowledgment should have been left in for transparency, unless it is incorrect.”
> > > > >
> > > > > Previously, we revised the original description mentioned by the reviewer because some reviewers misunderstood it as saying that we also exclude the *feature preprocessing steps* (which would result in OOD inputs to the ICL models), which is not true. We believe that the current version makes explicit that we
> > > > > - only disable PHE and *not* the internal feature preprocessing steps; and
> > > > > - use the “default” inference configuration for TabPFN-v2 (i.e., no additional per-dataset tuning).
> > > > >
> > > > > **References:**
> > > > >
> > > > > [1] TabArena: A Living Benchmark for Machine Learning on Tabular Data (Erickson et al., 2025)
> > > > >
> > > > > [2] TabICL: A Tabular Foundation Model for In-Context Learning on Tabular Data (Qu et al., 2025)
> > > > >
> > > > > [3] TabDPT: Scaling Tabular Foundation Models on Real Data (Ma et al., 2025)
> > > > >
> > > > > [4] Accurate Predictions on Small Data with A Tabular Foundation Model (Hollmann et al., 2025)

---

### Official Review · Reviewer_GPxJ · 2025-11-01

**Soundness:** 3
**Presentation:** 2
**Contribution:** 1
**Rating:** 2
**Confidence:** 4

**Summary:**

The paper empirically evaluates accuracy and generalisation properties of leading tabular foundation models. Authors curate 88 classification and 82 regression datasets and show that in some settings the FMs do not outperform classic baselines such as XGBoost.

**Strengths:**

Authors conduct a large and comprehensive empirical comparison, benchmarking leading TFMs and baselines. Multiple generalisation settings are evaluated including full training data and few shot. Some conclusions are important for future work and fair evaluation of TMFs, in particular the difference between statistics significance in the min-max/rank setting vs original space. The evaluation suite is publicly released.

**Weaknesses:**

Authors make a number of design choices that are not well justified such as z-score normalisation, mean/mode imputation, and removing regression target transforms. Furthermore, internal feature pre-processing steps for leading TFMs are also shut off to make to comparison "fair". I don't see how this makes the comparison fair, if a TFM is pre-trained with these feature transforms then shutting them off during inference will inevitably hurt performance. Instead, I think the same transforms should be made available to all baselines and swept over as part of the hyper-parameter search.

Paper also make a number of strong statements that are not well justified such as "simply scaling pretraining
over a diverse collection of tabular datasets may offer limited performance benefits". From Figure 1b, TabPFN-v2 still has double the win rate over the leading CatBoost baseline *without* re-training where as CatBoost is trained from scratch and heavily tuned on each dataset. I think the generalisation benefits of pre-training are very clear even under this evaluation setting.

Overall, while the paper shows some interesting findings, I think the experimental settings disadvantage TFMs and overly strong conclusions are drawn from the results.

**Questions:**

Do you have results when feature transforms are turned on (and swept over) for TFMs? These transforms can't be shut off if models were trained with them.

---

> ### Author Response · Authors · 2025-11-25
> **Author Response [1/2]**
>
> Thank you for your insightful comments and thoughtful feedback. We are encouraged to see that the reviewer found our work as presenting “conclusions [that] are important for future work and fair evaluation of TFMs, in particular the difference between statistics significance in the min-max/rank setting vs. original space” based on “a large and comprehensive comparison” that considers “multiple generalisation settings”. Below, we provide our response to the individual questions and concerns raised by the reviewer.
>
> > “internal feature pre-processing steps for leading TFMs are shut off to make to comparison “fair”. I don’t see how this makes the comparison fair, if a TFM is pre-trained with these feature transforms then shutting them off during inference will inevitably hurt performance.”
>
> > “I think the experimental settings disadvantage TFMs”
>
> > “Do you have results when feature transforms are turned on (and swept over) for TFMs? These transforms can’t be shut off if models were trained with them.”
>
> We would like to clarify a possible misunderstanding. We *do not* turn off the *feature preprocessing steps* considered by the tabular ICL models (TabPFN-v2, TabICL, TabDPT), which are the only TFMs with built-in feature preprocessing.
>
> More precisely:
> - TabPFN-v2: We use the “default” feature preprocessing steps recommended by the authors in Extended Data Table 5 of [1], which are internal to the model’s forward pass.
> - TabDPT: We use the same feature preprocessing steps provided by the authors in Appendix C.1 of [2].
> - TabICL: We use the same feature preprocessing steps provided by the authors in Section 5.1 of [3].
>
> These preprocessing steps are *internally* applied (i.e., within the forward pass of each ICL model), on top of the data that has already been preprocessed as detailed in the “Datasets” paragraph of Section 3.
>
> **As such, all of the TFM results (in particular, for the tabular ICL models) presented in the paper have all of the original feature transforms enabled.** The only part that we turn off is the post-hoc ensembling approach proposed for TabPFN-v2, which involves ensembling *multiple* TabPFN-v2 models as opposed to using just one instantiation of the model.
>
> To avoid any confusion, in our revised draft, we have updated the “Tabular foundation models (TFMs)” paragraph in Section 3 (**in blue**) to:
>
> “TabPFN-v2 can be evaluated with *post-hoc ensembling* (PHE; detailed in the “TabPFN (PHE)” section of Hollmann et al. (2025)), where each prediction is obtained by aggregating the outputs from *multiple* instantiations of TabPFN-v2 with greedy ensemble selection (Caruana et al., 2004). However, we *intentionally exclude PHE* for TabPFN-v2 to (i) ensure a consistent setup with other TFMs which are not post-hoc ensemble in the original paper and (ii) isolate the impact of tabular pretraining on downstream performance as much as possible. **The “default” preprocessing steps and inference configuration for TabPFN-v2 (see Extended Data Table 5 of Hollmann et al. (2025)) are still internally applied within the model’s forward pass.**”
>
> > “Paper… makes a number of strong statements that are not well justified such as “simply scaling pretraining over a diverse collection of tabular datasets may offer limited performance benefits.”
>
> > “overly strong conclusions are drawn from the results”
>
> First of all, we do not intend to claim that “tabular pretraining can never work”, i.e., that there is an inherent ceiling to the performance benefits from tabular pretraining. We also do not intend to suggest that *all* TFMs fail to show improvements, and believe that any observable improvement at the individual model level is transparently discussed throughout the paper (e.g., TabDPT on few-shot classification and regression).
>
> Rather, our main takeaway is that the positive conclusions of prior works on TFMs (i.e., “tabular pretraining leads to substantial performance gains”) may be *overly optimistic*. It is certainly possible that, at even larger scales and with methodological innovations for tabular pretraining, TFMs consistently show significant performance improvements over non-pretrained baselines, which we explicitly acknowledge in our Limitations section (Appendix E, **in orange**).
>
> Meanwhile, in the revised draft, we have updated the relevant text (**in red**) in the Abstract, Introduction (Section 1), and the Conclusion (Section 5) sections to:
>
> “Our findings suggest that current approaches to large-scale tabular pretraining may offer limited performance benefits, showing room for improvement in methods for effective tabular transfer learning.”

---

> > ### Author Response · Authors · 2025-11-25
> > **Author Response [2/2]**
> >
> > > “From Figure 1b, TabPFN-v2 still has double the win rate over… CatBoost *without* re-training where as CatBoost is… heavily tuned on each dataset. I think the generalisation benefits of pre-training are very clear even under this setting.”
> >
> > As noted in our response above, our intention is *not* to suggest that *all* TFMs fail to show any improvement. Rather, our main point in Figure 1 is that min-max normalization can amplify the *perceived* performance differences across models, often leading to overly optimistic interpretations (Figure 1(a)) than what raw performance with statistical testing would suggest (Figure 1(b)).
> >
> > For the setup considered in Figure 1, *after min-max normalization*, we observe that TabPFN-v2 achieves:
> > - An average (min-max normalized) AUROC of 0.9035 vs. 0.7777 for CB (Figure 1(a), left);
> > - A win rate of 75.9% and loss rate of 24.1% against CB (Figure 1(a), right).
> >
> > In contrast, when we compare them on the *original scale* and perform statistical testing (i.e., account for statistical “ties”), we find that TabPFN-v2 achieves:
> > - An average (unnormalized) AUROC of 0.9323 vs. 0.9213 for CB (Figure 1(b), left);
> > - A win rate of 20.7%, **a tie rate of 69%**, and a loss rate of 10.3% against CB (Figure 1(b), right).
> >
> > The results in the latter case are indeed still in favor of TabPFN-v2 but are far less pronounced than in the former, hence our characterization of the improvements as “modest” (or limited).
> >
> > We also note that in Finding 1 (Section 3, lines 313–315), we explicitly state the following **in bold** regarding these results:
> >
> > “These results are still in favor of TabPFN-v2 over CB, but the perceived gains are far less pronounced than in the former, showing “modest” improvements.”
> >
> > > “Authors make a number of design choices that are not well justified such as z-score normalisation, mean/mode imputation, and removing regression target transforms.”
> >
> > We first clarify that both z-score normalization (of numerical features) and mean imputation are commonly used in many prior works on tabular ML. Below, we list examples of prior works that use these preprocessing techniques, along with references to relevant sections:
> >
> > **Z-score normalization:**
> > - TabPFN-v2 [1]: “Details on the neural architecture” in Methods section
> > - TabDPT [2]: Appendix C.1
> > - XTab [6]: “Data pre-processing” in Section 4
> > - Schwartz-Ziv and Armon [7]: Section 3.1.1
> >
> > **Mean imputation:**
> > - TabPFN-v2 [1]: “Details on the neural architecture” in Methods section
> > - TabDPT [2]: Appendix C.1
> > - XTab [6]: “Data pre-processing” in Section 4
> > - TabZilla [4]: Appendix B.4
> >
> > We also only perform imputation if a given model is incapable of natively handling missing values (lines 182–183 of Section 3). For instance, models like TabuLa-8B (and its base model Llama-3-8B) and GBDT baselines (CatBoost, XGBoost, LightGBM) natively handle missing values, and we do not perform any imputation beforehand. All of the ICL TFMs—TabPFN-v2, TabDPT, and TabICL—internally perform mean imputation to handle missing values.
> >
> > Meanwhile, we provide additional results with the regression target transforms enabled (z-score normalized, as considered in e.g., [1–3,8]). We focus on TabPFN-v2 and TabDPT, which overall perform the best among TFMs on the regression tasks we consider.
> >
> > Below, we show the win/tie/loss rates of the two models against ModernNCA on the small regression datasets, *with and without regression target transforms*. We compare against ModernNCA, as it had the highest Elo rating among baselines in this data regime.
> >
> > Win/tie/loss Rates *without* regression target transforms (shown in Figure 2, top right):
> > | | Win | Tie | Loss |
> > | - | - | - | - |
> > | TabPFN-v2 | 22.2% | 52.8% | 25.0% |
> > | TabDPT | 19.4% | 63.9% | 16.7% |
> >
> > Win/tie/loss Rates *with* regression target transforms:
> > | | Win | Tie | Loss |
> > | - | - | - | - |
> > | TabPFN-v2 | 18.9% | 59.5% | 21.6% |
> > | TabDPT | 21.6% | 62.2% | 16.2% |
> >
> > We observe that our originally observed trends are relatively stable with respect to the presence/absence of regression target transforms. We therefore believe that our main findings are robust to these choices and remain valid.

---

> ### Author Response · Authors · 2025-11-25
> **Author Response [References Discussed in Above Responses]**
>
> **References:**
>
> [1] Accurate Predictions on Small Data with a Tabular Foundation Model (Hollmann et al., 2025)
>
> [2] TabDPT: Scaling Tabular Foundation Models on Real Data (Ma et al., 2025)
>
> [3] TabICL: A Tabular Foundation Model for In-Context Learning on Large Data (Qu et al., 2025)
>
> [4] When Do Neural Nets Outperform Boosted Trees on Tabular Data? (McElfresh et al., 2023)
>
> [5] CARTE: Pretraining and Transfer for Tabular Learning (Kim et al., 2024)
>
> [6] XTab: Cross-table Pretraining for Tabular Transformers (Zhu et al., 2023)
>
> [7] Tabular Data: Deep Learning is Not All You Need (Shwartz-Ziv and Armon, 2021)
>
> [8] Better by Default: Strong Pre-Tuned MLPs and Boosted Trees on Tabular Data (Holzmüller et al., 2024)

---

> > ### Comment · Area_Chair_UGxi · 2025-11-28
> > **Reminder to Respond to Rebuttal**
> >
> > Reviewer GPxJ,
> >
> > Please ensure that you read the authors' response to your review and post an acknowledgement **before end of Dec. 2**. This date is the cutoff for reviewer posts.
> >
> > Thank you for supporting quality peer review at ICLR.
> >
> > AC

---

### Official Review · Reviewer_VFzw · 2025-11-02

**Soundness:** 2
**Presentation:** 2
**Contribution:** 1
**Rating:** 2
**Confidence:** 4

**Summary:**

The paper evaluates seven open-source tabular foundation models (TFMs) on 88 classification and 82 regression datasets, in both full-data and few-shot regimes, to check whether the performance gains reported in prior work actually hold under stricter evaluation. It shows that common practices—min–max normalizing metrics and only doing rank-based tests—can overstate the benefits of tabular pretraining; when performance is kept on the original scale and per-dataset bootstrap significance is applied, TFMs are usually tied with strong tree/NN baselines like CatBoost, only clearly winning on small classification tables.

The authors also compare three TFMs (TabuLa-8B, TP-BERTa, XTab) with their non-pretrained counterparts and find that pretraining mainly helps LLM-style tabular ICL, but brings little or no benefit to other architectures, so the overall gains from large-scale tabular pretraining are modest.

**Strengths:**

1. Clear problem framing. The paper explicitly asks: “Do reported gains from tabular pretraining really hold up under stricter, per-dataset evaluation?”—a useful and timely question for the community.

2. Evaluation is careful and more realistic. Using original-scale metrics + per-dataset bootstrap CIs + win/tie/loss tallies is stricter than most prior TFM papers and exposes overclaiming.

3. Solid experiments covering sufficient classification and regression datasets.

**Weaknesses:**

1. Current TFMs are not a faithful proxy for “large-scale pretraining” on tabular data. The paper’s main takeaway is that existing tabular foundation models (TFMs) do not clearly outperform the best overall (highest Elo) baselines. However, this does not necessarily imply that the learning paradigm of large-scale pretraining “fails” on tabular data. A more cautious interpretation is that the current crop of TFMs is not yet able to exploit large-scale pretraining as effectively as models in vision or NLP. This gap may be attributable to architectural and methodological limitations rather than to an inherent ceiling of tabular pretraining itself.

2. The paper acknowledges successes of pretraining in CV and NLP, but it does not sufficiently connect these successes to the fact that those domains benefit from strong, well-understood inductive biases (locality, translation equivariance, token order, etc.). In tabular learning, such inductive biases are notably weaker or dataset-specific, making it harder for current TFMs to turn scale into generalization. For instance, TabPFN-v2 can be viewed as a relatively brute-force design that attends over samples and features via full self-attention without structural pruning, whereas in vision and language we routinely inject structure (e.g. shifted windows in Swin-Transformer, sparse / block attention in many LLMs). The paper does not explore whether better inductive structure—rather than “more data”—would change the outcome.

3. The paper reads more like a performance audit than a path forward. The four findings are largely descriptive—they carefully document that the reported advantages of TFMs shrink when evaluated per-dataset with proper uncertainty estimates. This is valuable, but the paper stops one step early: it does not turn these observations into concrete design guidance on how to build TFMs that actually benefit from scale on tables.

4. In the head-to-head comparison (TabuLa-8B, TP-BERTa, XTab vs. their non-pretrained counterparts), the paper concludes that pretraining mainly helps LLM-style models on in-context tabular tasks, but not other models. However, XTab does not actually support in-context learning in the same sense: it is pretrained across many datasets but still requires fine-tuning, and its featurizers and projection heads are data-specific, with only the transformer backbone being transferable. Putting XTab in the same “ICL-capable TFM” bucket risks conflating “multi-dataset pretraining” with “true ICL (train+query in the prompt)”, and thus weakens the generality of the claim in Finding 4.

**Questions:**

See weaknesses

---

> ### Author Response · Authors · 2025-11-25
> **Author Response [1/2]**
>
> Thank you for your insightful comments and constructive feedback. We are encouraged to see that the reviewer found our work as having a “clear problem framing” and as addressing a “useful and timely question for the community” by performing “careful and more realistic” and “valuable” evaluation that “exposes overclaiming”, with “solid experiments covering sufficient classification and regression datasets”. Below, we provide our response to the individual questions and concerns raised by the reviewer.
>
> > “Current TFMs are not a faithful proxy for “large-scale pretraining” on tabular data. The paper’s main takeaway… does not necessarily imply that the learning paradigm of large-scale pretraining “fails” on tabular data. A more cautious interpretation is that the current crop of TFMs is not yet able to exploit large-scale pretraining as effectively as models in vision or NLP. This gap may be attributable to architectural and methodological limitations rather than to an inherent ceiling of tabular pretraining itself.”
>
> > “The paper acknowledges success of pretraining in CV and NLP, but it does not sufficiently connect these successes to the fact that those domains benefit from strong, well-understood inductive biases… In tabular learning, such inductive biases are notably weaker or dataset-specific, making it harder for current TFMs to turn scale into generalization.”
>
> First of all, we do not intend to claim that “tabular pretraining can never work”, i.e., that there is an inherent ceiling to the performance benefits from tabular pretraining. While there are differences between the tabular and vision/language domains that may make tabular pretraining less effective (e.g., as you note, the weakness of inductive biases with tabular data), we agree that one cannot “prove the negative”, i.e., that pretraining will “never work” on tabular data, just on the basis of current models.
>
> Rather, we critically evaluate the widespread belief that “tabular pretraining currently works well”. All prior works on TFMs claim substantial improvements over non-pretrained baselines, while our findings suggest that these claims might be *overly optimistic*. Resolving this discrepancy remains valuable regardless of why current TFMs underperform (whether due to architectural shortcomings and/or the inherent heterogeneity of tabular data).
>
> To make these points clearer, in the revised draft, we have updated the relevant text (**in red**) in the Abstract, Introduction (Section 1), and the Conclusion (Section 5) sections to:
>
> “Our findings suggest that current approaches to large-scale tabular pretraining may offer limited performance benefits, showing room for improvement in methods for effective tabular transfer learning.”
>
> > “The paper does not explore whether better inductive structure—rather than “more data”—would change the outcome.”
>
> > “...reads more like a performance audit than a path forward… they carefully document that the reported advantages of TFMs shrink when evaluated per-dataset with proper uncertainty estimates. This is valuable, but the paper stop one step early: it does not turn these observations into concrete design guidance on how to build TFMs that actually benefit from scale on tables.”
>
> We believe that proposing a new TFM architecture that benefits from large-scale pretraining is beyond the scope of our paper, and that this does not detract from the value of our work.
>
> More reliable benchmarking approaches, like ours, are a prerequisite for assessing progress on making tabular pretraining more effective. We do not propose a new method for building TFMs, but rather a more thorough methodology for evaluating whatever the next generation of TFM approaches may be (e.g., with better inductive biases for effective cross-table transfer).

---

> ### Author Response · Authors · 2025-11-25
> **Author Response [2/2]**
>
> > “In the head-to-head comparison… XTab does not actually support in-context learning in the same sense… Putting XTab in the same “ICL-capable TFM” bucket risks conflating “multi-dataset pretraining” with “true ICL (train+query in the prompt)”, and thus weakens the generality of the claim in Finding 4.”
>
> We would like to clarify a possible misunderstanding. We do not claim that XTab is capable of ICL, nor do we place XTab in the same “ICL-capable TFM” bucket.
>
> What we actually claim in Finding 4 are:
> - Performance gains observed for TabuLa-8B vs. its non-pretrained counterpart (Llama-3-8B) suggest that tabular pretraining can improve the *tabular ICL* capabilities of *LLMs*.
> - Lack of performance gains for XTab and TP-BERTa vs. their non-pretrained counterparts suggests that despite being pretrained on other tabular datasets, they exhibit no clear advantage over the latter.
>
> In retrospect, this confusion may be due to the current (somewhat difficult to parse) title for Finding 4 in Section 3. In our revised draft, we have updated the title for Finding 4 (**in purple**) to read:
>
> “Head-to-head comparisons of three TFMs vs. their non-pretrained counterparts show that tabular pretraining (i) improves LLMs’ tabular ICL capabilities, (ii) but offers no clear performance benefits for other models, yielding limited overall gains (Figure 4).”

---

> > ### Comment · Area_Chair_UGxi · 2025-11-28
> > **Reminder to Respond to Rebuttal**
> >
> > Reviewer VFzw,
> >
> > Please ensure that you read the authors' response to your review and post an acknowledgement **before end of Dec. 2**. This date is the cutoff for reviewer posts.
> >
> > Thank you for supporting quality peer review at ICLR.
> >
> > AC

---

### Author Response · Authors · 2025-11-25
**General Response**

We thank all reviewers for their insightful comments and constructive feedback. We are encouraged to see that many reviewers found our work as addressing a “useful and timely question for the community” (VFzw), conducting “a wide range of interesting experiments” (Uc6Q) that are “well designed” (97fx), “careful and realistic” (VFzw), and “comprehensive” (GPxJ)”, and presenting conclusions that are “original” (97fx), “valuable” (VFzw), “important for future work and fair evaluation of TFMs” (GPxJ), and demonstrative of “common shortcoming[s] of the tabular modeling literature” (UC6Q). We are also glad to see that some reviewers found our work to have a “clear problem framing” (VFzw) and be “well written and very clear throughout” (97fx).

Below, we provide our response to the specific points raised by each reviewer as separate comments.

---

### Meta-Review · Area_Chair_CcVE · 2025-12-22

**Summary:**

The paper presents a large-scale evaluation of Tabular Foundation Models (TFMs) and argues that they offer "modest benefits" compared to baselines like CatBoost. The reviewers were split with scores of 2, 2, 6, and 6. After considering the feedback, I am inclined to recommend rejecting the paper because the experimental setup appears to disadvantage TFMs, leading to conclusions that may be too strong.

A major concern raised by reviewers is that the design choices seem to unfairly penalize the foundation models. For instance, the authors removed regression target transforms and enforced specific imputation methods. More importantly, reviewers noted that "shutting off" or altering internal feature pre-processing steps for TFMs to make the comparison "fair" is problematic. Since TFMs are pretrained with specific transforms, removing them during inference inevitably hurts their performance. A fairer approach would have been to allow the baselines to use these transforms as part of their hyperparameter search, rather than restricting the TFMs.

Furthermore, the comparison overlooks the practical utility of models like TabPFN. The baselines were heavily tuned (trained from scratch on each dataset), whereas TabPFN yields predictions in a single forward pass without retraining. Despite this, the paper's own results (Figure 1b) show that TabPFN-v2 still has double the win rate of the leading CatBoost baseline. This suggests that the generalization benefits of pretraining are actually quite clear, contradicting the paper's claim that there are "limited performance benefits."

Finally, the paper’s argument relies heavily on evaluating models on the "original scale" of the data. However, unnormalized tabular data is notoriously difficult to learn from due to varying ranges and units. Normalization is often a necessary step for unified representation learning.

Overall, while the paper offers some interesting findings, the experimental settings seem to handicap the TFMs, and the negative conclusions drawn are not fully supported by the data.

**Reviewer Concerns:**

See Summary.

**Reviewer Scores:**

See Summary.

---

### Decision · Program_Chairs · 2026-01-26

Reject